# Steering 3D Molecule Generation in Data-Sparse Regions via Distributional Physical Priors

## Abstract

Can we train a 3D molecule generator using data from dense regions to generate samples in sparse regions? This challenge can be framed as an out-of-distribution (OOD) generation problem. Existing works on OOD generation primarily focus on property shifts. However, the distribution shifts may come from structural variations in molecules, such as certain types of scaffolds, dubbed as physical priors. This work introduces a novel and principled diffusion-based generative framework, termed Geometric OOD Diffusion Model *(GODD)*, which enables training a generator on data-abundant distributions to generalize to data-scarce distributions under structure shifts. Specifically, we propose utilizing a designated equivariant asymmetric autoencoder to capture distributional physical priors. The asymmetric module allows generalization to unseen, out-of-distribution structural variations. As these captured physical priors represent distinct distributions, they can steer the generation of samples that are not in dense regions. We demonstrate that with these encoded structural-grained distributional physical priors, *GODD* does not need to train with any molecules from the sparse regions. We conduct extensive experiments across various out-of-distribution molecule generation tasks using benchmark datasets. Compared to alternative baselines, our approach shows a significant improvement of up to 65.6% in success rate, defined based on molecular validity, uniqueness, and novelty. Additionally, we show that our generative framework, steered by physical priors, can be readily adapted to canonical fragment-based drug design tasks, exhibiting promising performance.

## 1 Introduction

Geometric generative models are proposed to approximate the distribution of complex geometries and are used to generate feature-rich geometries (Watson et al., 2023; Xie et al., 2022). There has been fruitful research progress on 3D molecule generation based on geometric generative modeling. Recent representative models for generating 3D molecules in silicon include autoregressive (Luo & Ji, 2022), flow-based models (Garcia Satorras et al., 2021), and diffusion models (Hoogeboom et al., 2022). Among others, diffusion models have demonstrated their superior performance (Hoogeboom et al., 2022). However, these generative models require tremendous data to mimic the training distribution. They can barely generate samples that are rare or even absent in the training set, hindering their applicability to *de novo* molecule generation (Walters & Murcko, 2020).

Taking a canonical molecule dataset – QM9 as our running example, diverse scaffolds of molecules have varying proportions and frequencies in nature (Ramakrishnan et al., 2014; Wu et al., 2018). Our initial

Table 1: Preliminary results on QM9. In distribution, OOD I and OOD II encompass molecules with high-, low-, and rare-frequency scaffolds, respectively. Generated samples from EDM and GeoLDM, which are trained on molecules with source scaffolds, dominantly belong to the in-distribution scaffold set, indicating that they can only reflect the training data distribution.

| QM9 | Scaffold Propotion (%) | | |
|---|---|---|---|
| Domains | In-dist | OOD I | OOD II |
| # Molecules | 100,000 | 15,000 | 15,831 |
| # Scaffolds | 1,054 | 2,532 | 12,075 |
| Dataset | 76.4 | 11.5 | 12.1 |
| EDM | 91.4 | 2.7 | 4.9 |
| GeoLDM | 90.6 | 3.5 | 5.9 |

findings indicate that existing diffusion-based molecular generative models, such as EDM (Hoogeboom et al., 2022) and GeoLDM (Xu et al., 2023), effectively capture the training data distribution, generating molecules with high-frequency scaffolds. However, these models struggle to generate molecules with rare scaffolds (see Table 1). With the expressive power of state-of-the-art diffusion-based generators, we ask: *Can we train a diffusion model using data from dense regions to generate realistic and valid 3D samples in sparse regions?*

To address the data scarcity issue, we propose leveraging the concept of *out-of-distribution (OOD) generalization* and framing the problem as OOD generation. The intuition is that if we can train a model with a source data-dense region and it can generalize to new, desired distributions, then generating realistic and valid 3D molecules in data-sparse regions becomes feasible. Our objective, therefore, is to train a generator with data-abundant distribution and steer it to generate samples in sparse regions. The distribution shift generally comes from properties or core fragments, such as certain types of scaffolds or ring-structures (Wu et al., 2018; Zhuang et al., 2023). Certain sets of fragments or properties depict distinct distributions. Existing works on OOD generation mainly focus on property shifts (Lee et al., 2023; Klarner et al., 2024). They usually utilize a naive property predictor for guidance, where the properties are scalars. Due to the sparsity of the 3D fragments, it is imperative to design new OOD generative frameworks to deal with fragment shifts.

This paper introduces a novel and principled *GODD*, which utilizes the physical priors to steer the generation of 3D molecules in the data-sparse regions. The crux of enabling out-of-distribution generation under fragment shits is to learn generalizable and equivariant representations of the fragments inducing distribution shifts. The learned representations, a.k.a *distributional physical priors*, then are properly baked into the denoising process. Specifically, we leverage an asymmetric encoder-decoder architecture to characterize the physical priors, motivated by the success of asymmetric autoencoders in generalizable representation learning. This asymmetric design exhibits transferable learning capability across distributions, allowing for the generalization of unseen fragment variations, including out-of-distribution scaffolds or ring structures. In summary, our primary contributions are summarized as follows:

*First*, to the best of our knowledge, we are the first study to tackle 3D molecule generation in data-sparse regions and frame the problem as an out-of-distribution generation problem under fragment shift. We adopt the concept of asymmetric encoder-decoder to characterize the physical priors, which are used to steer the generation of valid 3D molecules in data-sparse regions. Moreover, We ensure and theoretically prove that the physical priors extracted by the designed asymmetric autoencoder are SE(3)-equivariant. Our proposed framework does not require additional training on OOD data.

*Second*, we evaluate out-of-distribution generation setting with benchmarking datasets. We compare it with alternative baselines, including vanilla generative models, such as EDM, GeoLDM, EquiFM, GeoBFN, and EEGSDE (Hoogeboom et al., 2022; Xu et al., 2023; Song et al., 2023a;b; Bao et al., 2023), and OOD generative models, including MOOD and CGD (Lee et al., 2023; Klarner et al., 2024). Besides, we empirically validate the effectiveness of asymmetric design in OOD generation with ablation studies. Extensive experimental results show that the physical priors enable the model to generate molecules with desired OOD fragment variations in data-sparse regions. The success rate of molecules generated by *GODD* is improved by up to 65.6% compared with existing methods.

*Third*, we demonstrate that our generative framework, guided by physical priors, can be applied to fragment-based OOD generation. We verify that our framework can be readily adapted to link multiple fragments under OOD settings. Specifically, we evaluated our method with a canonical fragment-based drug design task—linker design—and show that the proposed method exhibits promising performance in fragment linking within the OOD context (Igashov et al., 2024).

## 2 PROBLEM SETUP AND PRELIMINARIES

### 2.1 PROBLEM DEFINITION

**Notations:** Let $d$ be the dimensionality of node features; a 3D molecule can be represented as a point cloud denoted as $\mathcal{G} = \langle \mathbf{x}, \mathbf{h} \rangle$, where $\mathbf{x} = (\mathbf{x}_1, \ldots, \mathbf{x}_N) \in \mathbb{R}^{N \times 3}$ is the atom coordinate matrix and $\mathbf{h} = (\mathbf{h}_1, \ldots, \mathbf{h}_N) \in \mathbb{R}^{N \times d}$ is the node feature matrix containing atomic type, charge features,

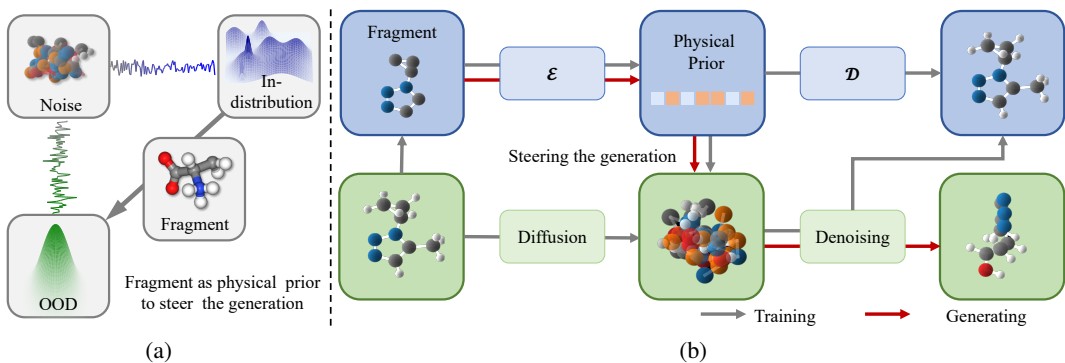

Figure 1: *The Illustration of Proposed* GODD *Framework.*
(a): *GODD* utilizes OOD fragments as physical priors to steer the generation toward data-sparse regions. (b): **During training (gray pipeline):** *I.* Encoder ($\mathcal{E}$) first maps fragments (i.e., scaffold/ring) into the latent features as physical priors. These latent features would be decoded ($\mathcal{D}$) for reconstructing the original molecule. This asymmetric encoder-decoder architecture enhances the generalization of representing unseen fragments for generating OOD samples; *II.* GODD first diffuses the molecule into noises and then utilizes physical priors to steer the denoising process toward molecules with given fragments. **During generation (red pipeline):** *GODD* receives the OOD fragment and encodes it as the physical prior. Then, the model denoises from sampled Gaussian noise under the guidance of physical prior, thereby generating novel and valid molecules with target fragment variations.

etc. For a given molecule $\mathcal{G}$, the fragment is a subgraph of the original molecule, represented as $\mathcal{G}^f = \langle \mathbf{x}^f, \mathbf{h}^f \rangle$. Specifically, the scaffold is its structural framework (Bemis & Murcko, 1996), termed as "chemotypes". Except for scaffolds, the ring structures are also essential fragments in chemistry and biology (Karageorgis et al., 2014; Ward & Beswick, 2014; Ritchie & Macdonald, 2009), which could also be a factor that incurs the distribution shift.

**Out-of-Distribution (OOD) Generation Problem:** We consider the problem of out-of-distribution generation in the following two scenarios: ODD scaffold and OOD ring-structure generation, respectively. Given a collection of molecules as training samples and corresponding in distributional fragment set (including scaffold or ring-structure) denoted as $\{\mathcal{G}_I\}$, $\{\mathcal{G}_I^f\}$, respectively. OOD generation aims to learn a generative model that can generate valid and novel molecules falling into a new distribution, where the corresponding fragment set is $\{\mathcal{G}_O^f\}$, and the OOD fragment set is unseen during training, a.k.a. $\{\mathcal{G}_I^f\} \cap \{\mathcal{G}_O^f\} = \emptyset$. We briefly review fragment-based drug design and OOD generation in Appendix L.

## 2.2 PRELIMINARIES

**Diffusion Models.** Diffusion models (Sohl-Dickstein et al., 2015) are latent variable models for learning distributions by modeling the reverse of a diffusion process (Ho et al., 2020). Given a data point $\mathbf{x}_0 \sim q(\mathbf{x}_0)$ and a variance schedule $\beta_1, \dots, \beta_T$ that controls the amount of noise added at each timestep $t$, the diffusion process or forward process gradually add Gaussian noise to the data point $\mathbf{x}$:

$$q(\mathbf{x}_t|\mathbf{x}_{t-1}) := \mathcal{N}(\mathbf{x}_t; \sqrt{1-\beta_t}\mathbf{x}_{t-1}, \beta_t\mathbf{I}). \tag{1}$$

Generally, the diffusion process $q$ has no trainable parameters. The denoising process or reverse process aims at learning a parameterized generative process, which incrementally denoise the noisy variables $\mathbf{x}_{T:1}$ to approximately restore the data point $\mathbf{x}_0$ in the original data distribution:

$$p_\theta(\mathbf{x}_{t-1}|\mathbf{x}_t) := \mathcal{N}(\mathbf{x}_{t-1}; \mu_\theta(\mathbf{x}_t, t), \mathbf{\Sigma}_\theta(\mathbf{x}_t, t)), \tag{2}$$

where the initial distribution $p(\mathbf{x}_t)$ is sampled from standard Gaussian noise $\mathcal{N}(0, \mathbf{I})$. The loss for training diffusion model $\mathcal{L}_{\text{DM}} := \mathcal{L}_t$ is simplified as: $\mathcal{L}_{\text{DM}} = \mathbb{E}_{\mathbf{x}_0, \epsilon, t}\left[\|\epsilon - \epsilon_\theta(\mathbf{x}_t, t)\|^2\right]$, where $w(t) = \frac{\beta_t}{2\sigma_t^2\alpha_t(1-\bar{\alpha}_t)}$ is the reweighting term and could be set as 1 with promising sampling quality, and $\mathbf{x}_t = \sqrt{\bar{\alpha}_t}\mathbf{x}_0 + \sqrt{1-\bar{\alpha}_t}\epsilon$. We provide a detailed description of diffusion models in Appendix A.

## 3 METHOD

**Overview.** Our objective is to train a generator with rich in distribution data that can be steered to a new distribution in a low-data regime. Generally, fragment variations, such as scaffold or ring-structure variations, are the main cause of the distribution shift in the context of OOD molecule generation (Ramakrishnan et al., 2014). We particularly focus on the geometric OOD generation problem where in distribution scaffold/ring-structure set, represented as $\{\mathcal{G}_I^f\}$, and the OOD scaffold/ring-structure set, denoted as $\{\mathcal{G}_O^f\}$, are different. In other words, the OOD scaffold/ring-structure set is unseen during training — $\{\mathcal{G}_I^f\} \cap \{\mathcal{G}_O^f\} = \emptyset$.

With the superior capability of diffusion models for 3D molecule generation, we propose to address the geometric OOD molecule generation problem with a diffusion engine. However, as illustrated in Section 1, the vanilla diffusion models or OOD methods have difficulty generating OOD molecules under fragment shifts. In this regard, we propose to incorporate the in-distribution fragments into the denoising process during training and the OOD ones into the denoising during generation. These fragments are learned as physical priors to steer the generation. Nevertheless, characterizing the physical priors that can transfer to new distributions is challenging because the OOD fragments are not seen during training. Inspired by the impressive generalizability of asymmetric autoencoder in both vision and language fields (He et al., 2022; Hu et al., 2022), we adopt an asymmetric encoder-decoder architecture to capture the physical priors in training distribution and to generalize to unseen OOD fragments. The proposed *GODD* workflow is provided in Figure 1.

### 3.1 EQUIVARIANT ASYMMETRIC AUTOENCODER

**Distributional Physical Prior.** For a given fragment $\mathcal{G}^f = \langle \mathbf{x}^f, \mathbf{h}^f \rangle$, the distributional physical prior learned from the fragment ($\mathcal{F}$) is defined as $\mathcal{F} = \langle \mathbf{f_x}, \mathbf{f_h} \rangle$. In the case of scaffold and ring-structure OOD generation, the fragments are atoms on the scaffold/rings.

**Asymmetric Autoencoder.** The asymmetric autoencoder comprises an encoder $\mathcal{E}$, which maps fragment $\mathcal{G}^f$ to a latent space, represented as $\mathbf{f_x}, \mathbf{f_h} = \mathcal{E}(\mathbf{x}^f, \mathbf{h}^f)$. Additionally, it includes a decoder $\mathcal{D}$ that reconstructs the latent representation back to the original molecular space, denoted as $\hat{\mathbf{x}}, \hat{\mathbf{h}} = \mathcal{D}(\mathbf{f_x}, \mathbf{f_h})$. Our autoencoder reconstructs the input by predicting the coordinates and features of complete atoms. The loss function computes the mean squared error (MSE) between the reconstructed and original molecules in the original molecular space. The autoencoder can be trained by minimizing the reconstruction objective, expressed as $\boldsymbol{f}(\mathcal{G}, \mathcal{D}(\mathcal{E}(\mathcal{G}^f)))$. The encoder of the autoencoder functions solely on the fragment $\mathcal{G}^f$, while the decoder reconstructs the input from the latent representation to the complete molecule $\mathcal{G}$. This asymmetric encoder-decoder design offers promising generalization (He et al., 2022) to the latent features. These features serve as physical prior and empower the model to generate molecules with unseen fragments.

**Equivariant Asymmetric Autoencoder.** However, naively applying autoencoder in the geometric domain is non-trivial. The diffusion model within the overall framework operates in 3D molecular space and necessitates conditions to be either equivariant or invariant. Therefore, it is crucial to ensure the equivariance of the conditions extracted by the autoencoder. To achieve this, we design our asymmetric autoencoder based on the Equivariant Graph Neural Networks (EGNNs) (Satorras et al., 2021), thereby incorporating equivariance into both the encoder $\mathcal{E}_\phi$ and decoder $\mathcal{D}_\vartheta$, where $\phi$ and $\vartheta$ are two learnable EGNNs. equivariant design ensures that the latent representations $\mathbf{f_x}$ and $\mathbf{f_x}$ encoded by the encoder from fragments are 3-D equivariant and $k$-d invariant, respectively. Consequently, Equivariant Asymmetric Autoencoder (EAAE) extracts both invariant and equivariant conditions, as expressed below:

$$\mathbf{R}\mathbf{f_x} + \boldsymbol{t}, \mathbf{f_h} = \mathcal{E}_\phi(\mathbf{R}\mathbf{x}^f + \boldsymbol{t}, \mathbf{h}^f) \tag{3}$$

$$\mathbf{R}\hat{\mathbf{x}} + \boldsymbol{t}, \hat{\mathbf{h}} = \mathcal{D}_\vartheta(\mathbf{R}\mathbf{f_x} + \boldsymbol{t}, \mathbf{f_h}), \tag{4}$$

for all rotations $\mathbf{R}$ and translations $\mathbf{t}$. Detailed architecture information about the asymmetric autoencoder can be found in Appendix B. The point-wise latent space adheres to the inherent structure of geometries $\mathcal{G}^f$, which facilitates learning conditions for the diffusion model and results in high-quality molecule design.

Following (Hoogeboom et al., 2022), to ensure that linear subspaces with the center of gravity always being zero can induce translation-invariant distributions, we define distributions of fragments $\mathbf{x}^f$,

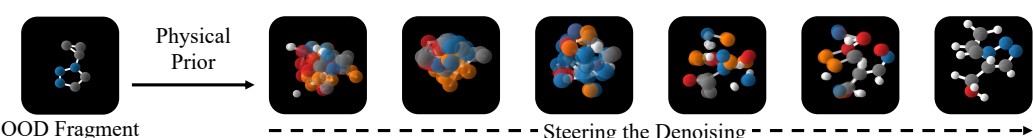

Figure 2: *The Illustration of Generating OOD Samples with* GODD: given an OOD fragment as the physical prior, our trained *GODD* can generate valid, unique, and novel molecules containing the target fragment.

physical priors $\mathbf{f}_x$, and reconstructed $\hat{\mathbf{x}}$ on the subspace that $\sum_i \mathbf{x}_i^f$ (or $\mathbf{f}_{x,i}$ and $\hat{\mathbf{x}}_i$) $= 0$. Then the encoding and decoding processes can be formulated by $q_\phi(\mathbf{f_x}, \mathbf{f_h}|\mathbf{x}^f, \mathbf{h}^f) = \mathcal{N}(\mathcal{E}_\phi(\mathbf{x}^f, \mathbf{h}^f), \sigma_0 \boldsymbol{I})$ and $p_\vartheta(\mathbf{x}, \mathbf{h}|\mathbf{f_x}, \mathbf{f_h}) = \prod_{i=1}^{N} p_\vartheta(x_i, h_i|\mathbf{f_x}, \mathbf{f_h})$ and the EAAE can be optimized by:

$$\mathcal{L}_{\text{EAAE}}(\mathcal{G}, \mathcal{G}^f) = \mathbb{E}_{q_\phi(\mathbf{f_x}, \mathbf{f_h}|\mathbf{x}^f, \mathbf{h}^f)} p_\vartheta(\mathbf{x}, \mathbf{h}|\mathbf{f_x}, \mathbf{f_h}) - \text{KL}[q_\phi(\mathbf{f_x}, \mathbf{f_h}|\mathbf{x}^f, \mathbf{h}^f) || \prod_i^{N} \mathcal{N}(f_{\mathbf{x},i}, f_{\mathbf{h},i}|0, \mathbf{I})], \tag{5}$$

where $\mathbb{E}_{q_\phi(\mathbf{f_x}, \mathbf{f_h}|\mathbf{x}^f, \mathbf{h}^f)} p_\vartheta(\mathbf{x}, \mathbf{h}|\mathbf{f_x}, \mathbf{f_h})$ is the asymmetric reconstruction loss and is calculated as $L_2$ norm or cross-entropy for continuous or discrete features. $\text{KL}[q_\phi(\mathbf{f_x}, \mathbf{f_h}|\mathbf{x}^f, \mathbf{h}^f) || \prod_i^{N} \mathcal{N}(f_x, f_h|0, \mathbf{I})]$ is a regularization term between $q_\phi$ and standard Gaussians. $\mathcal{L}_{\text{EAAE}}$ is standard VAE loss and is the variational lower bound of log-likelihood. The equivariance of the loss, which is crucial for geometric graph generation, is expressed as follows:

**Theorem 3.1.** $\mathcal{L}_{EAAE}$ *is an* $SE(3)$-*invariant variational lower bound to the log-likelihood, i.e., for any fragment* $\langle \mathbf{x}^f, \mathbf{h}^f \rangle$ *and molecule* $\langle \mathbf{x}, \mathbf{h} \rangle$, *we have* $\forall$ $\mathbf{R}$ *and* $\mathbf{t}$, $\mathcal{L}_{EAAE}(\mathbf{x}, \mathbf{h}, \mathbf{x}^f, \mathbf{h}^f) = \mathcal{L}_{EAAE}(\mathbf{Rx} + \mathbf{t}, \mathbf{h}, \mathbf{Rx}^f + \mathbf{t}, \mathbf{h}^f)$.

The theorem ensures that the asymmetric autoencoder is equivariant so that the extracted condition satisfies the equivariant constraints, thereby ensuring that the conditional denoising of the geometric diffusion model is also equivariant. Detailed proof of Theorem 3.1 is given in Appendix C. In summary, EAAE first inputs the physical prior $\mathcal{G}^f$ into the encoder $\mathcal{E}$ to obtain equivariant latent features $\mathbf{f_x}$ and invariant latent features $\mathbf{f_h}$. These features have two purposes. One is to continue to be input into the decoder $\mathcal{D}$ for reconstruction to constrain the latent features. Secondly, it is used as the condition to supervise and control the diffusion model. The specific method of the second part will be explained in the following section.

## 3.2 PHYSICAL PRIOR STEERED DIFFUSION MODEL

With the equivariant latent features $\langle \mathbf{f_x}, \mathbf{f_h} \rangle$, now we can utilize these features as domain supervisors for reconstructing structures $\mathcal{G}$ while still keeping geometric properties. The latent features encoded by the asymmetric encoder from the same molecule serve as the condition for the diffusion model. Such a similar manner to self-supervised learning enables the model to generate molecules with target structural variations, and thereby, the proposed method can perform adaptive molecule generation.

Generally, geometric diffusion models are capable of controllable generation with given conditions $s$ by modeling conditional distributions $p(\mathbf{z}|s)$. This modeling in DMs can be implemented with conditional denoising networks $\epsilon_\theta(\mathbf{z}, t, s)$ with the critical difference that it takes additional inputs $s$. However, an underlying constraint of such use is the assumption that $s$ is invariant. By contrast, a fundamental challenge for our method is that the conditions for the DM contain not only invariant features $\mathbf{f_h}$ but also equivariant features $\mathbf{f_x}$. This requires the distribution $p_\theta(\mathbf{z}_{0:T})$ of our DMs to satisfy the critical invariance:

$$\forall \mathbf{R}, \ p_\theta(\mathbf{z_x}, \mathbf{z_h}, \mathbf{f_x}, \mathbf{f_h}) = p_\theta(\mathbf{Rz_x}, \mathbf{z_h}, \mathbf{Rf_x}, \mathbf{f_h}), \tag{6}$$

where $\mathbf{z_x}$ and $\mathbf{z_h}$ are the noises. To achieve this, we should ensure that (1) the initial distribution $p(\mathbf{z}_{\mathbf{x},T}, \mathbf{z}_{\mathbf{h},T}, \mathbf{f_x}, \mathbf{f_h})$ is invariant, which is already satisfied since $\mathbf{z}_{\mathbf{x},T}$ is projected down by subtracting its center of gravity after sampling from standard Gaussian noise. With the $\mathbf{f_x}, \mathbf{f_h}$ is obtained by equivariant $\mathcal{E}_\phi$ (Equations 3); (2) the conditional reverse processes via $\theta$, which is expressed as $p_\theta(\mathbf{z}_{\mathbf{x},t-1}, \mathbf{z}_{\mathbf{h},t-1}|\mathbf{z}_{\mathbf{x},t}, \mathbf{z}_{\mathbf{h},t}, \mathbf{f_x}, \mathbf{f_h})$, are equivariant:

$$\forall \mathbf{R}, \ p_\theta(\mathbf{z}_{\mathbf{x},t-1}, \mathbf{z}_{\mathbf{h},t-1}|\mathbf{z}_{\mathbf{x},t}, \mathbf{z}_{\mathbf{h},t}, \mathbf{f_x}, \mathbf{f_h}) = p_\theta(\mathbf{Rz}_{\mathbf{x},t-1}, \mathbf{z}_{\mathbf{h},t-1}, |\mathbf{Rz}_{\mathbf{x},t}, \mathbf{z}_{\mathbf{h},t}, \mathbf{Rf_x}, \mathbf{f_h}), \tag{7}$$

this can be realized by implementing the denoising network $\epsilon_\theta$ with EGNN that satisfy the following equivariance:

$$\forall \, \mathbf{R} \text{ and } \mathbf{t}, \; \mathbf{R}\mathbf{z}_{\mathbf{x},t-1} + \mathbf{t}, \mathbf{z}_{\mathbf{h},t-1} = \epsilon_\theta(\mathbf{R}\mathbf{z}_{\mathbf{x},t} + \mathbf{t}, \mathbf{z}_{\mathbf{h},t}, \mathbf{R}\mathbf{f}_{\mathbf{x}} + \mathbf{t}, \mathbf{f}_{\mathbf{h}}, t), \tag{8}$$

To keep translation invariance, all the intermediate states $\mathbf{z}_{\mathbf{x},t}, \mathbf{z}_{\mathbf{h},t}$ are also required to lie on the subspace by $\sum_i \mathbf{z}_{\mathbf{x},t,i} = 0$ by moving the center of gravity. Analogous to Equation 17, now we can train the Physical Prior Steered Diffusion Model (PSDM) by:

$$\mathcal{L}_{\text{PSDM}}(\mathcal{G}, \mathcal{G}^f) = \mathbb{E}_{\mathcal{G}, \mathcal{E}(\mathcal{G}^f), \epsilon, t} \left[ \|\epsilon - \epsilon_\theta(\mathbf{z}_{\mathbf{x},t}, \mathbf{z}_{\mathbf{h},t}, \mathbf{f}_{\mathbf{x}}, \mathbf{f}_{\mathbf{h}}, t)\|^2 \right] \tag{9}$$

with $w(t)$ simply set as 1 for all steps t. As the EGNN only receives atomic coordinates and features $\mathbf{z}_{\mathbf{x},t}$ and $\mathbf{z}_{\mathbf{h},t}$, we concatenate $\mathbf{f}_{\mathbf{x}}$ and $\mathbf{f}_{\mathbf{h}}$ to the node features $\mathbf{z}_{\mathbf{h},t}$. Specifically, with node features $\mathbf{z}_{\mathbf{h},t} \in \mathbb{R}^{N \times d}$, a time-step embedding $\mathbf{t} \in \mathbb{R}^{N \times 1}$, $\mathbf{f}_{\mathbf{x}} \in \mathbb{R}^{N' \times 3}$, and $\mathbf{f}_{\mathbf{h}} \in \mathbb{R}^{N' \times k}$, the EGNN within the denoising network $\epsilon_\theta$ processes coordinates $\mathbf{z}_{\mathbf{x},t} \in \mathbb{R}^{N \times 3}$ and concatenated features $\mathbf{z}_{\mathbf{h},t} \in \mathbb{R}^{N \times (d+3+k+1)}$. Since the number of fragments $N'$ is less than the number of molecules $N$, zeros are padded to $\mathbf{f}_{\mathbf{x}}$ and $\mathbf{f}_{\mathbf{h}}$.

### 3.3 TRAINING AND GENERATING OOD SAMPLES

**Training.** The training loss of the entire framework can be formulated as $\mathcal{L} = \mathcal{L}_{\text{EAAE}} + \mathcal{L}_{\text{PSDM}}$. To make the training loss tractable, we also show that $\mathcal{L}$ is theoretically an SE(3)-invariant variational lower bound of the log-likelihood, and we can have:

**Theorem 3.2.** *Let $\mathcal{L} := \mathcal{L}_{EAAE} + \mathcal{L}_{PSDM}$. With certain weights $w(t)$, $\mathcal{L}$ is an $SE(3)$-invariant variational lower bound to the log-likelihood.*

Given the above training loss and Theorem 3.2, we can optimize *GODD* via back-propagation with reparameterizing trick (Kingma & Welling, 2013). We provide the detailed proof of Theorem 3.2 in Appendix D, and a formal description of the optimization procedure in Algorithm 1 in Appendix F. We follow the process of EDM (Hoogeboom et al., 2022) regarding the representation for continuous features $\mathbf{x}$ and categorical features $\mathbf{h}$. For clarity, we provided the details in Appendix B.3.

**Generating OOD Molecules.** With *GODD* trained on dataset $\{\mathcal{G}_I\}$ and given an OOD scaffold/ring-structure $\mathcal{G}_O^f$, we can perform OOD molecule generation (a scaffold OOD generative process is illustrated in Figure 2). To sample from the model, one first inputs the $\mathcal{G}_O^f$ into the encoder $\mathcal{E}_\phi$ and obtains the latent representation of $\mathcal{G}_O^f$ denoted as physical prior $\langle \mathbf{f}_{\mathbf{x}}, \mathbf{f}_{\mathbf{h}} \rangle$ via reparameterization. With the OOD physical prior as condition, the framework first samples $\mathbf{z}_{\mathbf{x},T}, \mathbf{z}_{\mathbf{h},T} \sim \mathcal{N}_{x,h}(\mathbf{0}, \mathbf{I})$ and then iteratively samples $\mathbf{z}_{\mathbf{x},t-1}, \mathbf{z}_{\mathbf{h},t-1} \sim p_\theta(\mathbf{z}_{\mathbf{x},t-1}, \mathbf{z}_{\mathbf{h},t-1} | \mathbf{z}_{\mathbf{x},t}, \mathbf{z}_{\mathbf{h},t}, \mathbf{f}_{\mathbf{x}}, \mathbf{f}_{\mathbf{h}})$. Finally, the output molecule represented as $\langle \mathbf{x}, \mathbf{h} \rangle$ is sampled from $p(\mathbf{z}_{\mathbf{x},0}, \mathbf{z}_{\mathbf{h},0} | \mathbf{z}_{\mathbf{x},1}, \mathbf{z}_{\mathbf{h},1}, \mathbf{f}_{\mathbf{x}}, \mathbf{f}_{\mathbf{h}})$. The pseudo-code of the adaptive generation is provided in Algorithm 2 in Appendix F.

## 4 EXPERIMENTS

### 4.1 EXPERIMENT SETUP

**Datasets and Tasks.** We evaluate over QM9 (Ramakrishnan et al., 2014) and the GEOM-DRUG (Axelrod & Gómez-Bombarelli, 2022). Specifically, QM9 is a standard dataset that contains molecular properties and atom coordinates for 130k 3D molecules with up to 9 heavy atoms and up to 29 atoms, including hydrogens. GEOM-DRUG encompasses around 450,000 molecules, each with an average of 44 atoms and a maximum of 181. Dataset details and experimental parameters are presented in Appendices G, H, and E.

*Ring-Structure Molecule Generation.* In this task, ring-structure variations result in distribution shifts. We used RDKit (Landrum et al., 2016) to categorize molecules into nine groups based on the number of rings, ranging from 0 to 8. As the number of rings increases, the quantity of molecules correspondingly decreases. We partition the QM9 dataset into two subsets based on ring count. The training data distribution comprises molecules and those with 0 to 3 rings, and we consider the five target distributions including molecules with 4 to 8 rings, respectively. Figure 6 in the Appendix presents a schematic diagram illustrating example molecules with 0 to 8 rings. The GEOM-DRUG

dataset contains molecules with 0 to 14 rings and 22 rings. We include molecules with 0 to 10 rings as the training set and consider five target distributions as the number of molecules with 11 to 14 and 22 rings are all under 100, representing data-sparse regions.

*Scaffold Molecule Generation.* In this task, scaffold variations lead to distribution shifts. We used RDKit (Landrum et al., 2016) to examine the scaffold of each molecule in the QM9 dataset. Molecules without a scaffold were marked as '-' and included in the total scaffold count. The dataset was divided based on scaffold frequency. Specifically, the in-distribution dataset contained 100,000 molecules and 1,054 scaffolds, with most scaffolds appearing at least 100 times. Out-of-distribution I included 15,000 molecules and 2,532 scaffolds, where most scaffolds appeared between 10 to 100 times. Out-of-distribution II consisted of 15,831 molecules and 12,075 scaffolds, with each scaffold appearing less than 10 times. Our goal is to train a generative model using the in-distribution data to generate effective molecules that fall into desired new distributions, such as OOD I and II.

*Linker Design.* The proposed method, leveraging the target fragment to steer the generation towards data-sparse regions, fundamentally falls into the paradigm of fragment-based drug design (Murray & Rees, 2009). In addition to the aforementioned tasks, we extend our framework to linker design and demonstrate a proof-of-concept of *GODD* on canonical fragment-based design tasks under the OOD settings. In particular, we observe that the GEOM-LINKER dataset exhibits fragment shifts due to the ring number of molecules, with molecules having a ring number above eight being extremely sparse. For comparisons, we split the GEOM-LINKER according to the number of rings and included molecules with sparse ring numbers as the OOD dataset for testing. Further details about the GEOM-LINKER dataset and related works are provided in Appendices I and L.

**Baselines.** To comprehensively compare performance, we include unconditional, conditional, and OOD generative frameworks. First, we employ four state-of-the-art 3D unconditional molecule diffusion models: EDM (Hoogeboom et al., 2022), GeoLDM (Xu et al., 2023), EquiFM (Song et al., 2023a), and GeoBFN (Song et al., 2023b), to validate the efficacy of our proposed *GODD* in OOD generation. Second, we apply EEGSDE (Bao et al., 2023) and modify EDM and GeoLDM for conditional generation. As these methods can only control the generation process with scalar features, we use the number of rings as a scalar feature in ring-structure molecule generation. We set ring counts as the condition to control the generation process of the baselines, denoted as C-EDM, C-GeoLDM, and EEGSDE, to verify *GODD*'s effectiveness in the OOD ring-structure generation task. Lastly, we include OOD generative frameworks, including MOOD (Lee et al., 2023) and CGD (Klarner et al., 2024), for ring-structure molecule generation to compare the performance of OOD generation. For comparative purposes, we also train unconditional models on the entire dataset (denoted with †) and highlight models trained exclusively on in-distribution data with ‡.

For linker design, we will use DiffLinker (Igashov et al., 2024) and LinkerNet (Guan et al., 2024) as the baselines for comparisons. DiffLinker developed a diffusion model capable of connecting multiple molecular fragments, while LinkerNet further advanced this by introducing diffusion models on Riemann manifolds for fragment linking.

**Metrics.** Our objective is to generate effective 3D molecules in data-sparse regions. A generated sample is effective only when it falls into the target distribution while it is valid, unique, and novel simultaneously. Therefore, our evaluation metrics can be defined as follows:

1. **Proportion (P)**: Given an OOD scaffold/ring set $\{\mathcal{G}_O^f\}$, proportion describes the percentage of molecules that contain the desired scaffold/ring-structure in $\{\mathcal{G}_O^f\}$ among generated valid samples; 2. **Coverage (C)**: Coverage describes the percentage of scaffold $\{\mathcal{G}_O^f\}$ set of the generated samples (denoted as $\{\mathcal{G}_G^f\}$) in the ODD scaffold set $\{\mathcal{G}_O^f\}$, which is expressed as $C = |\{\mathcal{G}_G^f\}|/|\{\mathcal{G}_O^f\}|$; 3. **Target atom stability (AS)**: The ratio of atoms that has the correct valency with the desired scaffold/ring-structure among all generated molecules; 4. **Target molecule stability (MS)**: The ratio of generated molecules contains the desired scaffold/ring-structure, and all atoms are stable. GEOM-DRUG dataset has nearly 0% molecule-level stability, so this metric is generally ignored on GEOM-DRUG (Hoogeboom et al., 2022); 5. **Target validity (V)**: The percentage of valid molecules among all the desired molecules, which is measured by RDkit (Landrum et al., 2016) and widely used for calculating validity (Hoogeboom et al., 2022; Xu et al., 2023)); 6. **Target novelty (N)**: The percentage of novel molecules among all the desired valid molecules, the novel molecule is different from training samples; 7. **Success rate (S)**: The ratio of generated valid, unique, and novel molecules that contain the desired scaffold/ring-structure.

Table 2: Results of molecule proportion in terms of ring-number (P), atom stability (AS), molecule stability (MS), validity (V), novelty (N), and success rate (S). The **best** results are highlighted in bold. QM9 contains 36 eight-ring molecules, and the proportion is nearly 0.

| Metrics ↑ | P (%) in Distribution | | | | P (%) in OOD Generation | | | | | AS | MS | V | N | S |
|---|---|---|---|---|---|---|---|---|---|---|---|---|---|---|
| No. of Ring | 0 | 1 | 2 | 3 | 4 | 5 | 6 | 7 | 8 | Averaged metrics (%) | | | | |
| QM9 | 10.2 | 39.3 | 27.6 | 15.1 | 4.4 | 2.7 | 0.6 | 0.2 | 0.0 | 99.0 | 95.2 | 97.7 | - | - |
| EDM† | 10.5 | 39.8 | 28.0 | 14.5 | 4.0 | 2.9 | 0.2 | 0.1 | 0.0 | 11.0 | 9.6 | 10.4 | 6.8 | 6.3 |
| GeoLDM† | 12.0 | 38.6 | 27.0 | 15.3 | 4.6 | 2.2 | 0.2 | 0.1 | 0.0 | 11.0 | 9.9 | 10.4 | 6.4 | 5.9 |
| EDM‡ | 12.1 | 44.1 | 29.8 | 11.8 | 1.7 | 0.5 | 0.0 | 0.0 | 0.0 | 11.0 | 9.7 | 10.4 | 6.8 | 6.3 |
| GeoLDM‡ | 2.8 | 41.5 | 32.1 | 15.7 | 4.7 | 2.7 | 0.3 | 0.1 | 0.0 | 10.9 | 9.1 | 10.4 | 6.7 | 6.2 |
| EquiFM‡ | 3.5 | 41.9 | 32.6 | 15.0 | 4.6 | 2.3 | 0.0 | 0.0 | 0.0 | 11.0 | 9.8 | 10.5 | 6.0 | 5.6 |
| GeoBFN‡ | 3.6 | 41.7 | 32.5 | 15.5 | 4.6 | 2.1 | 0.0 | 0.0 | 0.0 | 11.0 | 10.1 | 10.6 | 7.4 | 7.0 |
| C-EDM‡ | 98.9 | 94.2 | 80.8 | 64.4 | 12.6 | 26.8 | 0.3 | 0.1 | 0.0 | 41.3 | 33.9 | 38.0 | 27.3 | 24.1 |
| C-GeoLDM‡ | 97.1 | 89.4 | 74.2 | 52.4 | 22.3 | 22.7 | 0.9 | 0.2 | 0.0 | 39.1 | 31.5 | 35.7 | 28.3 | 25.0 |
| EEGSDE‡ | 98.4 | 92.2 | 77.6 | 58.2 | 14.1 | 17.6 | 0.3 | 0.0 | 0.0 | 39.1 | 31.1 | 35.7 | 27.2 | 24.2 |
| MOOD‡ | 80.7 | 87.1 | 86.1 | 73.3 | 34.1 | 32.3 | 10.3 | 0.2 | 0.0 | 44.3 | 39.0 | 42.1 | 25.5 | 21.0 |
| CGD‡ | 82.3 | 84.8 | 86.2 | 83.6 | 34.4 | 22.4 | 10.3 | 10.1 | 0.0 | 45.5 | 40.0 | 43.2 | 28.4 | 26.2 |
| *GODD*‡ | **99.9** | **99.8** | **99.1** | **97.6** | **92.5** | **89.7** | **78.7** | **88.2** | **82.1** | **83.1** | **54.0** | **77.9** | **70.3** | **40.5** |

†: Models are trained over entire QM9;
‡: Models are trained over ring-split QM9 with ring-number from 0-3.
C-: C-EDM and C-GeoLDM are trained with conditioning on ring counts.

## 4.2 RESULTS AND ANALYSIS

**Ring-Structure Molecule Generation.** In this task, all models were trained with the same training data that contains molecules with ring counts ranging from 0 to 3. Subsequently, their OOD generative performances were tested for generating molecules with 4 to 8 rings, respectively. We present the results on 10,000 generated molecules for each ring-count distribution in Table 2. For clarity, the generated target molecule validity, novelty, and success rate are calculated by averaging the corresponding values from 4 training distributions and 5 target distributions. Full results are presented in Appendix J.

Table 2 demonstrates that those uncontrollable methods baselines (i.e., EDM, GeoLDM, EquiFM, and GeoBFN) can barely generate molecules with 4 to 8 rings — with 7.0% success rate at most. Manipulating the generation process with ring counts can

Table 3: Results of molecule proportion in terms of ring number (P), atom stability (AS), molecule validity (V), novelty (N), and success rate (S). The number of molecules with above 11 rings in GEOM-DRUG is lower than 100.

| Method | Averaged Metric (%) ↑ | | | | |
|---|---|---|---|---|---|
| | P | AS | V | N | S |
| GEOM-DRUG | 0.0 | 86.5 | 99.9 | - | - |
| EDM† | 0.0 | 0.0 | 0.0 | 0.0 | 0.0 |
| GeoLDM† | 0.0 | 0.0 | 0.0 | 0.0 | 0.0 |
| EquiFM† | 0.0 | 0.0 | 0.0 | 0.0 | 0.0 |
| GeoBFN† | 0.0 | 0.0 | 0.0 | 0.0 | 0.0 |
| *GODD*‡ | **13.8** | **11.4** | **11.0** | **13.8** | **10.9** |

† Models are trained on complete GEOM-DRUG.
‡ Models are trained on GEOM-DRUG with ring numbers from 0-10.

slightly improve OOD generation performance with up to 25% success rates. OOD generative models show slight improvement but are still insignificant. In contrast, *GODD* can achieve a 40.5% success rate. Moreover, we observe that no baselines can generate 8-ring molecules, including those controllable generation methods (i.e., C-GeoLDM, C-EDM, and EEGSDE) and OOD methods (MOOD and CGD), reflecting the difficulty of generating those complex and sparse molecules in the original QM9 (only 36 8-ring molecules). Notably, *GODD* can generate 82.1% portion of 8-ring molecules even though the training data does not contain any of those samples, showing the significance of using physical prior representations for steering the denoising process of the diffusion models. Specifically, among the generated 10,000 molecules using *GODD*, 2,388 valid, unique, and novel 8-ring molecules were obtained. These results verify that *GODD* can perform OOD 3D molecule generation with the ring-structure shifts in data-sparse distributions.

Table 3 presented the statistical results of various methods for generating rare ring number molecules (ranging from 11 to 14 and 22) on the large-scale dataset GEOM-DRUG, in which the molecules with large ring numbers are even more sparse. Notably, EDM, GeoLDM, EquiFM, and GeoBFN, which are even trained on the complete dataset, cannot generate molecules with ring numbers exceeding 10, thus failing to produce any desired molecules. In contrast, *GODD* can generate an average of 13.8% of the OOD molecules by solely training on molecules with ring numbers from 0-10. Specifically, for molecules with 22 rings, of which there are only two in the complete dataset, *GODD* produces 1,374 valid and novel molecules out of 10,000 generated samples, whereas none

Table 4: Results of proportion (P), scaffold coverage (C), molecule validity (V), molecule novelty (N), and molecule success rate (S). The **best** results are highlighted in bold.

| Domains | In distribution (%) | | | | | OOD I (%) | | | | | OOD II (%) | | | | |
|---|---|---|---|---|---|---|---|---|---|---|---|---|---|---|---|
| Metrics↑ | P | C | V | N | S | P | C | V | N | S | P | C | V | N | S |
| Data | 76.4 | 100.0 | 97.7 | - | - | 11.5 | 100.0 | 97.7 | - | - | 12.1 | 100.0 | 97.7 | - | - |
| EDM† | 79.9 | 36.3 | 74.8 | 48.8 | 45.0 | 10.9 | 28.9 | 10.2 | 6.7 | 6.1 | 9.2 | 34.9 | 8.6 | 5.6 | 5.2 |
| GeoLDM† | 80.4 | 35.2 | 75.6 | 46.7 | 43.1 | 10.7 | 31.2 | 10.1 | 6.2 | 5.8 | 8.8 | 33.5 | 8.3 | 5.1 | 4.7 |
| EquiFM† | 80.4 | 36.8 | 76.1 | 43.2 | 40.9 | 7.8 | 35.1 | 7.3 | 4.2 | 3.9 | 11.8 | 29.2 | 0.0 | 0.0 | 0.0 |
| GeoBFN† | 81.3 | 35.2 | 77.5 | 54.0 | 51.4 | 7.7 | 34.3 | 7.4 | 5.1 | 4.9 | 11.0 | 32.0 | 0.0 | 0.0 | 0.0 |
| EDM‡ | 91.4 | 56.5 | 83.2 | 58.2 | 52.0 | 5.9 | 26.5 | 5.3 | 3.7 | 3.3 | 2.7 | 17.0 | 2.4 | 1.7 | 1.5 |
| GeoLDM‡ | 90.6 | 54.3 | 81.7 | 57.8 | 51.0 | 5.9 | 26.7 | 5.3 | 3.8 | 3.3 | 3.5 | 19.0 | 3.2 | 2.3 | 2.0 |
| EquiFM‡ | 91.0 | 56.3 | 86.2 | 48.9 | 46.3 | 5.4 | 27.8 | 5.1 | 2.9 | 2.7 | 3.6 | 17.4 | 0.0 | 0.0 | 0.0 |
| GeoBFN‡ | 91.1 | 54.4 | 86.8 | 60.5 | 57.7 | 6.0 | 27.3 | 5.7 | 4.0 | 3.8 | 2.9 | 19.9 | 2.7 | 1.9 | 1.8 |
| *GODD*‡ | **99.2** | **92.5** | **90.7** | **67.6** | **52.4** | **97.0** | **97.1** | **80.0** | **84.5** | **68.9** | **95.5** | **85.7** | **83.3** | **82.0** | **65.8** |

† Models are trained over the entire QM9 dataset;
‡ Models are trained only with in-distribution data, where each scaffold appears at least 100 times.

Table 5: Results of atom stability (AS) and molecule stability (MS). The **best** results are highlighted in bold.

| Domains | In-dist (%) | | OOD I (%) | | OOD II (%) | |
|---|---|---|---|---|---|---|
| Metrics↑ | AS | MS | AS | MS | AS | MS |
| Data | 99.0 | 95.2 | 99.0 | 95.2 | 99.0 | 95.2 |
| EDM† | 78.9 | 65.5 | 10.8 | 8.9 | 9.1 | 7.5 |
| GeoLDM† | 79.5 | 71.9 | 10.6 | 9.6 | 8.7 | 7.9 |
| EquiFM† | 79.5 | 71.0 | 6.3 | 6.0 | 0.0 | 0.0 |
| GeoBFN† | 80.5 | 73.9 | 7.3 | 7.0 | 0.0 | 0.0 |
| EDM‡ | 90.4 | 73.3 | 5.8 | 4.7 | 2.6 | 2.1 |
| GeoLDM‡ | 89.1 | 75.6 | 5.8 | 4.9 | 3.5 | 3.0 |
| EquiFM‡ | 90.0 | 80.4 | 5.3 | 4.8 | 3.6 | 3.2 |
| GeoBFN‡ | 90.3 | **82.8** | 5.9 | 5.5 | 2.9 | 2.6 |
| GODD‡ | **96.1** | 71.3 | **89.5** | **45.6** | **89.0** | **35.1** |

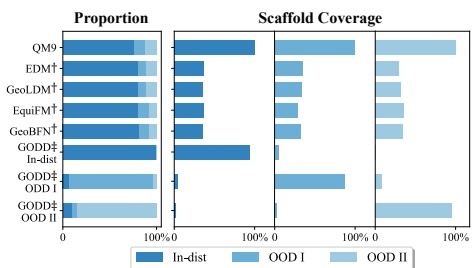

Figure 3: **Visualization of Proportion and Coverage.** Compared methods can only mimic the original distribution and are incapable of generating OOD molecules. Besides, only molecules generated by the proposed method cover OOD scaffolds.

of the compared methods can generate even a single molecule with 22 rings. The proposed method achieves a remarkable improvement in the success rate by 13.7% in generating such molecules, even without exposure to these two molecules.

**Scaffold Molecule Generation.** In the task of OOD scaffold molecule generation, the scaffolds are too sparse to train an effective classifier for guidance-based generative models (15,831 molecules contain 12,075 different scaffolds); we then train unconditional methods both on the complete dataset (†) and in-distribution data (‡) for a comprehensive comparison. In particular, our *GODD* is trained exclusively over the in-distribution dataset. After training, each model generates 15,000 molecules for the in-distribution, OOD I, and OOD II. The quantitative results using various metrics are presented in Table 4, Table 5, and Figure 3. We observe that with EDM, GeoLDM, EquiFM, and GeoBFN, the scaffold proportion of the generated molecules indeed mirrors that of the training samples (see proportion and coverage visualization in Figure 3). However, they all struggle to generate molecules with scaffolds falling into the desired distribution I or II; they can only achieve 3.8% success rates at most (see Table 4). In contrast, our proposed *GODD*, trained solely on the in-distribution data, can generate OOD molecules containing the target scaffolds given the corresponding fragments, achieving at least 95.5% proportion in both new distributions.

Notably, for OOD II, comprising over 12,000 different rare scaffolds, only *GODD* can achieve 85.7% coverage. Nevertheless, all baselines can only achieve 35.1% coverage at most, indicating the significance of our EAAE. It is worth noting that *GODD* does not require any OOD molecules; instead, it encodes the fragment as the physical prior for OOD generation, overcoming the data scarcity challenge. *GODD* improves the molecule novelty and success rate by up to 80.1% regarding novelty and 64.0% in terms of success rate as compared to the

Table 6: Results on the quantitative estimate of drug-likeness (QED), synthetic accessibility (SA), validity (v), and success rate (S) on the linker design task. The **best** results are highlighted in bold.

| GEOM-LINKER | QED↑ | SA↓ | V (%)↑ | S (%)↑ |
|---|---|---|---|---|
| DiffLinker | 0.56 | 3.92 | 42.17 | 14.45 |
| LinkerNet | 0.56 | 3.89 | 48.5 | 18.9 |
| *GODD* | **0.57** | **3.63** | **65.2** | **22.61** |

baselines. The atom stability and molecule stability presented in Table 5 also demonstrates that the designed *GODD* performs better on generating chemically stable molecules with desired scaffolds.

**Evaluation on the Task of Linker Design.** In addition to validity and uniqueness, we include metrics from previous works, such as the quantitative estimate of drug-likeness (QED) and synthetic accessibility (SA). The experimental results indicate that existing linker design methods fall short in linking OOD fragments, achieving a validity below 50%. In contrast, we can achieve a validity of 65.2%. These results demonstrate that *GODD* shows promising performance in fragment linking within the OOD context.

**Ablation Study for Evaluating the Significance of the Asymmetric Autoencoder.** We present the ablation study in Table 7 featuring a variation of the proposed method, *GODD*\*, which utilizes a *symmetric autoencoder*. Specifically, the autoencoder of *GODD*\* receives and reconstructs only the fragment. The results indicate that *GODD*\* demonstrates promising in-distribution generation

Table 7: Results of proportion (P), scaffold coverage (C), molecule validity (V), molecule success rate (S), atom stability (AS), and molecule stability (MS). The **best** results are highlighted in bold.

| Domains | In-dist (%) | | | OOD I (%) | | | OOD II (%) | | |
|---|---|---|---|---|---|---|---|---|---|
| Metrics↑ | P | C | V | P | C | V | P | C | V |
| *GODD*\* | **99.2** | **98.5** | 85.1 | 95.1 | 96.9 | 58.3 | 94.3 | 84.0 | 35.0 |
| *GODD*‡ | **99.2** | 92.5 | **90.7** | **97.0** | **97.1** | **80.0** | **95.5** | **85.7** | **83.3** |
| Metrics↑ | AS | MS | S | AS | MS | S | AS | MS | S |
| *GODD*\* | 89.2 | 68.4 | 52.1 | 82.0 | 12.8 | 41.8 | 75.1 | 10.4 | 31.0 |
| *GODD*‡ | **96.1** | **71.3** | **52.4** | **89.5** | **45.6** | **68.9** | **89.0** | **35.1** | **65.8** |

and achieves better performance in scaffold coverage, aligning with the performance of traditional autoencoders in the in-distribution tasks. However, *GODD*\* performs worse than *GODD* in OOD generation. Although *GODD*\* achieves similar proportions and coverage by receiving OOD fragments, its generation quality is worse, particularly regarding stability and validity. This suggests that even with fragments, *GODD*\* is still hard to generalize to generate valid molecules in OOD scenarios. These observations underscore the effectiveness of using asymmetric autoencoder.

**Limitations.** This paper addresses the problem of OOD generation in the context of structural shifts. However, in some scenarios, OOD structures may not be provided. We plan to investigate this issue in future work by developing methods to identify structural variations when OOD structures are unavailable. Additionally, most generative models, including ours, adopt the EGNN modules to capture the equivariance of molecules (Hoogeboom et al., 2022; Xu et al., 2023; Song et al., 2023a;b). The model's memory overhead escalates exponentially with the size of the input molecules, posing a significant constraint, especially for generating large molecules. Given a molecule $\mathcal{G} = \langle \mathbf{x} \in \mathbb{R}^{n \times 3}, \mathbf{h} \in \mathbb{R}^{n \times f} \rangle$. Suppose the total number of layers of EGNNs used is $l$ and the hidden feature for EGNN is $h$, then the space complexity of our model is $\mathcal{O}(nnhl)$. For example, in the GEOM-DRUG dataset, if molecules of 180 atoms are processed, all methods EGNN-based algorithms require around 3.5GB of memory, which results in huge overhead for experiments.

## 5 CONCLUSION

This paper investigated the problem of OOD molecule generation in the context of fragment shifts and proposed an asymmetric autoencoder to represent fragments as physical priors to steer the generation toward data-sparse regions. Our quantitative experiments demonstrated that the proposed method surpasses existing techniques, including unconditional, conditional, and OOD approaches, in generating valid, unique, and novel OOD molecules with desired fragments in data-sparse regions. Extensive quantitative results in successful OOD generation validated the ability of asymmetric autoencoder to encode unseen fragments and the potential of *GODD* in steering generation through the encoded physical priors. Furthermore, the linker design experiment confirmed the proposed method's applicability to fragment-based drug design. Additionally, our framework is generative model-agnostic; it can be seamlessly integrated into other generative models, such as latent diffusion (Xu et al., 2023) or flow-based models (Song et al., 2023a).

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

# APPENDIX

## A    DIFFUSION MODELS

Given a data point $\mathbf{x}_0 \sim q(\mathbf{x}_0)$ and a variance schedule $\beta_1, \ldots, \beta_T$ that controls the amount of noise added at each timestep $t$, the diffusion process or forward process gradually add Gaussian noise to the data point $\mathbf{x}$:

$$q(\mathbf{x}_t|\mathbf{x}_{t-1}) := \mathcal{N}(\mathbf{x}_t; \sqrt{1-\beta_t}\mathbf{x}_{t-1}, \beta_t\mathbf{I}), \tag{10}$$

where $\beta_{1:T}$ are chosen such that data point $\mathbf{x}$ will approximately converge to standard Gaussian, *i.e.*, $q(\mathbf{x}_T) \approx \mathcal{N}(0, \mathbf{I})$. Generally, the diffusion process $q$ has no trainable parameters. The denoising process or reverse process aims at learning a parameterized generative process, which incrementally denoise the noisy variables $\mathbf{x}_{T:1}$ to approximate restore the data point $\mathbf{x}_0$ in the original data distribution:

$$p_\theta(\mathbf{x}_{t-1}|\mathbf{x}_t) := \mathcal{N}(\mathbf{x}_{t-1}; \mu_\theta(\mathbf{x}_t, t), \mathbf{\Sigma}_\theta(\mathbf{x}_t, t)), \tag{11}$$

where the initial distribution $p(\mathbf{x}_t)$ is sampled from standard Gaussian noise $\mathcal{N}(0, \mathbf{I})$. The means $\mu_\theta$ typically are neural networks such as U-Nets for images or Transformers for text.

The forward process is $q(\mathbf{x}_{1:T}|\mathbf{x}_0)$ is an approximate posterior to the Markov chain, and the reverse process $p_\theta(\mathbf{x}_{0:T})$ is optimized by a variational lower bound on the negative log-likelihood of $\mathbf{x}_0$ by:

$$\mathbb{E}[-\log p_\theta(\mathbf{x}_0)] \le \mathbb{E}_q\left[-\log\frac{p_\theta(\mathbf{x}_{0:T})}{q(\mathbf{x}_{1:T}|\mathbf{x}_0)}\right] \tag{12}$$

$$= \mathbb{E}_q\left[-\log p(\mathbf{x}_T) - \sum_{t\ge 1}^{T}\log\frac{p_\theta(\mathbf{x}_{t-1}|\mathbf{x}_t)}{q(\mathbf{x}_t|\mathbf{x}_{t-1})}\right], \tag{13}$$

which is $\mathcal{L}_{\text{vlb}}$. To efficiently train the diffusion models, further improvements come to term $\mathcal{L}_{\text{vlb}}$ by variance reduction, and thereby Eq. (12) is rewritten as:

$$\mathcal{L}_{\text{vlb}} = \mathbb{E}_q[\mathcal{L}_T + \sum_{t=2}^{T}\mathcal{L}_t + \mathcal{L}_0] \tag{14}$$

where $\mathcal{L}_T = \log\frac{q(\mathbf{x}_T|\mathbf{x}_0)}{p_\theta(\mathbf{x}_T)}$, which models the distance between a standard normal distribution and the final latent variable $q(\mathbf{x}_T|\mathbf{x}_0)$, since the approximate posterior $q$ has no learnable parameters, so $\mathcal{L}_T$ is a constant during training and can be ignored. $\mathcal{L}_0 = -\log p_\theta(\mathbf{x}_0|\mathbf{x}_1)$ models the likelihood of the data given $\mathbf{x}_0$, which is close to zero and ignored as well if $\beta_0 \approx 0$ and $\mathbf{x}_0$ is discrete.

$\mathcal{L}_t$ in Eq. (14) is the loss for the reverse process and is given by:

$$\mathcal{L}_t = \sum_{t\ge 2}^{T}\log\frac{q(\mathbf{x}_{t-1}|\mathbf{x}_0, \mathbf{x}_t)}{p_\theta(\mathbf{x}_{t-1}|\mathbf{x}_t)}. \tag{15}$$

While in this formulation the neural network directly predicts $\hat{\mathbf{x}}_0$, (Ho et al., 2020) found that optimization is easier when predicting the Gaussian noise instead. Intuitively, the network is trying to predict which part of the observation $\mathbf{x}_t$ is noise originating from the diffusion process, and which part corresponds to the underlying data point $\mathbf{x}_0$. Then sampling $\mathbf{x}_{t-1} \sim p_\theta(\mathbf{x}_{t-1}|\mathbf{x}_t)$ is to compute

$$\mathbf{x}_{t-1} = \frac{1}{\sqrt{\alpha_t}}\left(\mathbf{x}_t - \frac{\sqrt{\beta_t}}{\sqrt{1-\bar{\alpha}_t}}\epsilon_\theta(\mathbf{x}_t, t)\right) + \sigma_t\mathbf{z}, \tag{16}$$

where $\alpha_t := 1 - \beta_t$, $\bar{\alpha}_t := \prod_{s=1}^{t}\alpha_s$, and $\mathbf{z} \sim \mathcal{N}(\mathbf{0}, \mathbf{I})$. And thereby $\mathcal{L}_{\text{DM}} := \mathcal{L}_t$ is simplified to:

$$\mathcal{L}_{\text{DM}} = \mathbb{E}_{\mathbf{x}_0, \epsilon, t}\left[w(t)\|\epsilon - \epsilon_\theta(\mathbf{x}_t, t)\|^2\right] \tag{17}$$

where $w(t) = \frac{\beta_t}{2\sigma_t^2\alpha_t(1-\bar{\alpha}_t)}$ is the reweighting term and could be simply set as 1 with promising sampling quality, and $\mathbf{x}_t = \sqrt{\bar{\alpha}_t}\mathbf{x}_0 + \sqrt{1-\bar{\alpha}_t}\epsilon$.

## B  MODEL ARCHITECTURE DETAILS

### B.1  EQUIVARAINT MASKED AUTOENCODER

In this work, EAAE considers visible molecular structural geometries as point clouds, without specifying the connecting bonds. Therefore, in practice, we take the point clouds as fully connected graph $\mathcal{G}$ and model the interactions between all atoms $v_i \in \mathcal{V}$. Each node $v_i$ is embedded with coordinates $\mathbf{x}_i \in \mathbb{R}^3$ and atomic features $\mathbf{h}_i \in \mathbf{R}^d$. Then, EAAE are composed of multiple Equivariant Convolutional Layers, and each single layer is expressed as (Satorras et al., 2021):

$$
\begin{aligned}
d_{ij}^2 &= \|\mathbf{x}_i^l - \mathbf{x}_j^l\|^2, \\
\mathbf{m}_{i,j} &= \phi_e(\mathbf{h}_i^l, \mathbf{h}_j^l, d_{ij}^2, a_{ij}), \\
\mathbf{x}_i^{l+1} &= \mathbf{x}_i^l + \sum_{j \neq i} \frac{\mathbf{x}_i^l - \mathbf{x}_j^l}{d_{ij} + 1} \phi_x(\mathbf{m}_{i,j}) \\
\mathbf{h}_i^{l+1} &= \phi_h(\mathbf{h}_i^l, \sum_{j \in \mathcal{N}(i)} \phi_i(\mathbf{m}_{ij})\mathbf{m}_{ij})
\end{aligned}
\tag{18}
$$

where $l$ denotes the layer index, $\phi_i(\mathbf{m}_{ij})$ reweights messages passed from different edges in an attention weights manner, $d_{ij} + 1$ is normalizing the relative directions $\mathbf{x}_i^l - \mathbf{x}_j^l$ following previous methods (Satorras et al., 2021; Hoogeboom et al., 2022). All learnable functions, *i.e.*, $\phi_e, \phi_x, \phi_h$, and, $\phi_i$, are parameterized by Multi Layer Perceptrons (MLPs). Then a complete EGNN model can be realized by stacking $L$ layers such that and satisfies the required equivariant constraint in Equations 3, 4, and 6.

### B.2  EQUIVARAINT PHYSICAL PRIOR STEERED DENOISING NEURAL NETWORKS

The denoising neural network is implemented by multiple equivariant convolutional layers, and the difference in the Equation 18 is the hidden features $\mathbf{h}$. Due to the diffusion model is conditioned by $\mathbf{f_x}, \mathbf{f_h}$ from encoder $\mathcal{E}$, the hidden features for our denoising neural network is expressed as $\bar{\mathbf{h}} \leftarrow [\mathbf{h}, \mathbf{f_x}, \mathbf{f_h}]$, where $\mathbf{h}$ are original features of geometric graph and $[a, b]$ is concatenation operation.

### B.3  MULTI-MODAL FEATURE REPRESENTATION OF MOLECULES

Multimodal features of molecules raise concerns for the term $\mathcal{L}_0 = -\log p_\theta(\mathbf{x}_0|\mathbf{x}_1)$ in Equation 14. For categorical features such as the atom types, this model would however introduce an undesired bias (Hoogeboom et al., 2022). For the intermediate variable $\mathbf{x}_t$, we subdivide it into $\mathbf{z}_{\mathbf{x},t}$ and $\mathbf{z}_{\mathbf{h},t}$ in the proposed DM, which are coordinate variables and atomic attribute variables, respectively.

**Coordinate Features.** First we set $\sigma_t^2\mathbf{I} \leftarrow \Sigma_\theta(\mathbf{x}_t, t) = \beta_t$ and add an additional correction term containing the estimated noise $\boldsymbol{\epsilon}_{\mathbf{x},0}$ from denoising neural network $\boldsymbol{\epsilon}$. Then continuous positions $\mathbf{z_x}$ in $p(\mathbf{z}_{\mathbf{x},0}|\mathbf{z}_{\mathbf{x},1})$ is expressed as:

$$
p(\mathbf{z}_{\mathbf{x},0}|\mathbf{z}_{\mathbf{x},1}) = \mathcal{N}(\mathbf{z}_{\mathbf{x},0}|\mathbf{z}_{\mathbf{x},1}/\alpha_1 - \sigma_1/\alpha_1\boldsymbol{\epsilon}_{\mathbf{x},0}, \sigma_1^2/\alpha_1^2\mathbf{I})
\tag{19}
$$

**Atom Type Features.** For categorical features such as the atom type, the aforementioned integer representation is unnatural and introduces bias. Instead of using integers for these features, we operate directly on a one-hot representation. Suppose $\mathbf{h}$ or $\mathbf{z}_{\mathbf{h},0}$ is an array whose values represent atom types in $\{c_1, \ldots, c_d\}$. Then $\mathbf{h}$ is encoded with a one-hot function $\mathbf{h} \leftarrow \mathbf{h}^{\text{one-hot}}$ such that $\mathbf{h}_{i,j}^{\text{one-hot}} \leftarrow \mathbf{1}_{h_i=c_i}$. and diffusion process over $\mathbf{z}_{\mathbf{h},t}$ at timestep $t$ and sampling at final timestep are given as:

$$
q(\mathbf{z}_{\mathbf{h},t}|\mathbf{z}_{\mathbf{h},0}) = \mathcal{N}(\mathbf{z}_{\mathbf{h},t}|\alpha_t\mathbf{h}^{\text{one-hot}}, \sigma_t^2\mathbf{I})
\tag{20}
$$

$$
p(\mathbf{z}_{\mathbf{h},0}|\mathbf{z}_{\mathbf{h},1}) = \mathcal{C}(\mathbf{z}_{\mathbf{h},0}|\mathbf{p}), \; \mathbf{p} \propto \int_{\mathbf{1}-\frac{1}{2}}^{\mathbf{1}+\frac{1}{2}} \mathcal{N}(\boldsymbol{u}; \mu_\theta(\mathbf{z}_{\mathbf{h},1}, 1), \sigma_1^2)\mathrm{d}\boldsymbol{u}
\tag{21}
$$

where $\mathbf{p}$ is normalized to sum to one and $\mathcal{C}$ is a categorical distribution.

**Atom Charge.** Atom charge is the ordinal type of physical quantity, and its sampling process at the final timestep can be formulated by standard practice (Ho et al., 2020):

$$p(\mathbf{z}_{\mathbf{h},0}|\mathbf{z}_{\mathbf{h},1}) = \int_{\mathbf{h}-\frac{1}{2}}^{\mathbf{h}+\frac{1}{2}} \mathcal{N}(\boldsymbol{u}; \mu_\theta(\mathbf{z}_{\mathbf{h},1}, 1), \sigma_1^2)\mathrm{d}\boldsymbol{u} \tag{22}$$

**Feature Scaling.** To normalize the features and make them easier to process for the neural network, we add weights to different modalities. The relative scaling has a deeper impact on the model: when the features $\mathbf{h}$ are defined on a smaller scale than the coordinates $\mathbf{x}$, the denoising process tends to first determine rough positions and decide on the atom types only afterward. Empirical knowledge shows that the weights for coordinate, atom type, and atom charge are 1, 0.25, and 0.1, respectively (Hoogeboom et al., 2022).

## C    LOSS OF EMAE IS SE(3)-INVARIANT

**Equivariance.** Molecules, typically existing within a three-dimensional physical space, are subject to geometric symmetries, including translations, rotations, and potential reflections. These are collectively referred to as the Euclidean group in 3 dimensions, denoted as E(3) (Celeghini et al., 1991). A function $F$ is said to be equivariant to the action of a group $G$ if $T_g \circ F(\mathbf{x}) = F \circ S_g(\mathbf{x})$ for all $g \in G$, where $S_g$, $T_g$ are linear representations related to the group element $g$ (Serre et al., 1977). We consider the special Euclidean group SE(3) for geometric graph generation involving translations and rotations. Moreover, the transformations $S_g$ or $T_g$ can be represented by a translation $\mathbf{t}$ and an orthogonal matrix rotation $\mathbf{R}$. For a molecule $\mathcal{G} = \langle \mathbf{x}, \mathbf{h} \rangle$, the node features $\mathbf{h}$ are SE(3)-invariant while the coordinates $\mathbf{x}$ are SE(3)-equivariant, which can be expressed as $\mathbf{R}\mathbf{x} + \mathbf{t} = (\mathbf{R}\mathbf{x}_1 + \mathbf{t}, \ldots, \mathbf{R}\mathbf{x}_N + \mathbf{t})$.

*Proof.* $\mathcal{L}_{\mathbf{EAAE}}$ **is** $SE(3)$**-invariance**

Recall the loss function:

$$\mathcal{L}_{\mathbf{EAAE}} = \mathbb{E}_{q_\phi(\mathbf{f_x},\mathbf{f_h}|\mathbf{x}^f,\mathbf{h}^f)} p_\vartheta(\mathbf{x},\mathbf{h}|\mathbf{f_x},\mathbf{f_h}) - \mathrm{KL}[q_\phi(\mathbf{f_x},\mathbf{f_h}|\mathbf{x}^f,\mathbf{h}^f)||\prod_i^N \mathcal{N}(f_{\mathbf{x},i}, f_{\mathbf{h},i}|0,\mathbf{I})] \quad (23)$$

Our expected outcome is $\forall \mathbf{R}$, $\mathcal{L}_{\mathbf{EAAE}}(\mathbf{x},\mathbf{h},\mathbf{x}^f,\mathbf{h}^f) = \mathcal{L}_{\mathbf{EAAE}}(\mathbf{R}\mathbf{x},\mathbf{h},\mathbf{R}\mathbf{x}^f,\mathbf{h}^f)$. We have:

$$\mathcal{L}_{\mathbf{EAAE}}(\mathbf{R}\mathbf{x},\mathbf{h},\mathbf{R}\mathbf{x}^f,\mathbf{h}^f)$$

$$= \mathbb{E}_{q_\phi(\mathbf{f_x},\mathbf{f_h}|\mathbf{R}\mathbf{x}^f,\mathbf{h}^f)} p_\vartheta(\mathbf{R}\mathbf{x},\mathbf{h}|\mathbf{f_x},\mathbf{f_h}) - \mathrm{KL}[q_\phi(\mathbf{f_x},\mathbf{f_h}|\mathbf{R}\mathbf{x}^f,\mathbf{h}^f)||\prod_i^N \mathcal{N}(f_{\mathbf{x},i}, f_{\mathbf{h},i}|0,\mathbf{I})]$$

$$= \int_\mathcal{G} q_\phi(\mathbf{f_x},\mathbf{f_h}|\mathbf{R}\mathbf{x}^f,\mathbf{h}^f) \log p_\vartheta(\mathbf{R}\mathbf{x},\mathbf{h}|\mathbf{f_x},\mathbf{f_h}) + \int_\mathcal{G} \log \frac{q_\phi(\mathbf{f_x},\mathbf{f_h}|\mathbf{R}\mathbf{x}^f,\mathbf{h}^f)}{\prod_i^N \mathcal{N}(f_{\mathbf{x},i}, f_{\mathbf{h},i}|0,\mathbf{I})}$$

$$= \int_\mathcal{G} q_\phi(\mathbf{R}\mathbf{R}^{-1}\mathbf{f_x},\mathbf{f_h}|\mathbf{R}\mathbf{x}^f,\mathbf{h}^f) \log p_\vartheta(\mathbf{R}\mathbf{x},\mathbf{h}|\mathbf{R}\mathbf{R}^{-1}\mathbf{f_x},\mathbf{f_h})$$

$$+ \int_\mathcal{G} \log \frac{q_\phi(\mathbf{R}\mathbf{R}^{-1}\mathbf{f_x},\mathbf{f_h}|\mathbf{R}\mathbf{x}^f,\mathbf{h}^f)}{\prod_i^N \mathcal{N}(f_{\mathbf{x},i}, f_{\mathbf{h},i}|0,\mathbf{I})} \qquad\qquad \mathbf{R}\mathbf{R}^{-1} = \mathbf{I}$$

$$= \int_\mathcal{G} q_\phi(\mathbf{R}^{-1}\mathbf{f_x},\mathbf{f_h}|\mathbf{x}^f,\mathbf{h}^f) \log p_\vartheta(\mathbf{x},\mathbf{h}|\mathbf{R}^{-1}\mathbf{f_x},\mathbf{f_h})$$

$$+ \int_\mathcal{G} \log \frac{q_\phi(\mathbf{R}^{-1}\mathbf{f_x},\mathbf{f_h}|\mathbf{x}^f,\mathbf{h}^f)}{\prod_i^N \mathcal{N}(f_{\mathbf{x},i}, f_{\mathbf{h},i}|0,\mathbf{I})} \qquad\qquad SE(3) \text{ of } \mathbf{x}, \mathbf{f_x}, \& \mathbf{x}^f$$

$$= \int_\mathcal{G} q_\phi(\mathbf{k},\mathbf{f_h}|\mathbf{x}^f,\mathbf{h}^f) \log p_\vartheta(\mathbf{x},\mathbf{h}|\mathbf{k},\mathbf{f_h}) \cdot |\mathbf{R}|$$

$$+ \int_\mathcal{G} \log \frac{q_\phi(\mathbf{k},\mathbf{f_h}|\mathbf{x}^f,\mathbf{h}^f)}{\prod_i^N \mathcal{N}(f_{\mathbf{x},i}, f_{\mathbf{h},i}|0,\mathbf{I})} \qquad\qquad \text{Let } \mathbf{k} = \mathbf{R}^{-1}\mathbf{f_x}$$

$$= \mathbb{E}_{q_\phi(\mathbf{k},\mathbf{f_h}|\mathbf{R}\mathbf{x}^f,\mathbf{h}^f)} p_\vartheta(\mathbf{x},\mathbf{h}|\mathbf{k},\mathbf{f_h})$$

$$- \mathrm{KL}[q_\phi(\mathbf{k},\mathbf{f_h}|\mathbf{x}^f,\mathbf{h}^f)||\prod_i^N \mathcal{N}(f_{\mathbf{x},i}, f_{\mathbf{h},i}|0,\mathbf{I})] \qquad\qquad |\mathbf{R}| = 1$$

$$= \mathcal{L}_{\mathbf{EAAE}}(\mathbf{x}^f,\mathbf{h}^f)$$

$$(24)$$

$$\square$$

Given the fragment $\mathcal{G}^f$, we subtract the center of gravity from $\mathbf{x}^f \in \mathcal{G}^f$, and thereby ensure that $\mathcal{E}$ receives isotropic geometric graph, and all together guarantee that the loss of EAAE is SE(3)-invariant.

# D LOSS OF *GODD* IS AN SE(3)-INVARIANT VARIATIONAL LOWER BOUND TO THE LOG-LIKELIHOOD

First, we present the rigorous statement of the Theorem 3.2 here:

**Theorem D.1.** *Given predefined and valid $\{\alpha_i\}_{i=0}^T$, $\{\beta_i\}_{i=0}^T$, and $\{\gamma_i\}_{i=0}^T$ Let $w(t)$ satisfies:*

$$1. \; \forall t \in [1, \ldots, T], w(t) = \frac{\beta_t^2}{2\gamma_t^2(1-\beta_t)(1-\alpha_t^2)} \tag{25}$$

$$2. \; w(0) = -1 \tag{26}$$

*Then given the geometric datapoint $\mathcal{G} = \langle \mathbf{x}, \mathbf{h} \rangle \in \mathbb{R}^{N \times (3+d)}$ and its subset $\mathcal{G}^f \langle \mathbf{x}^f, \mathbf{h}^f \rangle \in \mathbb{R}^{F \times (3+d)}$ the loss $\mathcal{L}$ of the proposed method is expressed as:*

$$\mathcal{L} := \mathcal{L}_{EAAE} + \mathcal{L}_{PSDM} \tag{27}$$

*which satisfies:*

$$1. \; \forall \mathbf{R} \text{ and } \mathbf{t}, \; \mathcal{L}(\mathbf{x}, \mathbf{h}, \mathbf{x}^f, \mathbf{h}^f) = \mathcal{L}(\mathbf{R}\mathbf{x} + \mathbf{t}, \mathbf{h}, \mathbf{R}\mathbf{x}^f + \mathbf{t}, \mathbf{h}^f) \tag{28}$$

$$2. \; \mathcal{L}(\mathbf{x}, \mathbf{h}, \mathbf{x}^f, \mathbf{h}^f) \geq -\mathbb{E}_{p_{\langle \mathbf{x}, \mathbf{h} \rangle \in \{\mathcal{G}\}, [\mathbf{f_x}, \mathbf{f_h}] = \mathcal{E}_\phi(\mathcal{G}^f)}} [\log p_\theta(\mathbf{z_x}, \mathbf{z_h} | \mathbf{f_x}, \mathbf{f_h})] \tag{29}$$

*And we have $\log p_\theta(\mathbf{x}_0, \mathbf{h}_0)$ is the marginal distribution of $\langle \mathbf{x}, \mathbf{h} \rangle$ under* GODD.

The theorem proposed herein posits two distinct assertions. Firstly, Equation 28 illustrates that the loss function $\mathcal{L}$ is $SE(3)$-invariant, meaning it remains unchanged under any rotational or translational transformations. Secondly, Equation 29 suggests that $\mathcal{L}$ acts as a variational lower bound for the log-likelihood. We provide comprehensive proofs for these assertions separately, commencing with Equation 29.

*Proof.* $\mathcal{L}$ **is a variational lower bound of the log-likelihood**

Recall the loss function:

$$\mathcal{L}(\mathbf{x}, \mathbf{h}, \mathbf{x}^f, \mathbf{h}^f) = \mathcal{L}_{\text{EAAE}} + \mathcal{L}_{\text{PSDM}} \tag{30}$$

$$= \mathbb{E}_{q_\phi(\mathbf{f_x}, \mathbf{f_h} | \mathbf{x}^f, \mathbf{h}^f)} p_\vartheta(\mathbf{x}, \mathbf{h} | \mathbf{f_x}, \mathbf{f_h}) - \text{KL}[q_\phi(\mathbf{f_x}, \mathbf{f_h} | \mathbf{x}^f, \mathbf{h}^f) \| \prod_i^N \mathcal{N}(f_{\mathbf{x}, i}, f_{\mathbf{h}, i} | 0, \mathbf{I})] \tag{31}$$

$$+ \mathbb{E}_{\mathcal{G}, \mathcal{E}_\phi(\mathcal{G}^f), \epsilon, t} \left[ \| \epsilon - \epsilon_\theta(\mathbf{x}_t, \mathbf{h}_t, \mathbf{f_x}, \mathbf{f_h}, t) \|^2 \right] \tag{32}$$

$\mathcal{L}_{\text{EAAE}}$ is a standard variational autoencoder and has been proven to be a variational lower bound of the log-likelihood (Kingma & Welling, 2014). For simplicity, we denote $\mathbf{z}_{\mathbf{x}, t}, \mathbf{z}_{\mathbf{h}, t}$ as $\mathbf{z}_t$, and $\mathbf{f_x}, \mathbf{f_h}$ as $\mathbf{f}$, then we expect $\mathcal{L}_{\text{PSDM}}$ has:

$$\log p_\theta(\mathbf{z} | \mathbf{f}) \geq \text{KL}[q(\mathbf{z}_{1:T} | \mathbf{z}_0) \| p_\theta(\mathbf{z} | \mathbf{f})] \tag{33}$$

$$
\begin{aligned}
\log p_\theta(\mathbf{z}|\mathbf{f}) \geq & \mathbb{E}_{q(\mathbf{z}_{1:T}|\mathbf{z}_0)}\left[\log \frac{p_\theta(\mathbf{z}_{0:T}|\mathbf{f})}{q(\mathbf{z}_{1:T}|\mathbf{z}_0)}\right] \\
= & \mathbb{E}_{q(\mathbf{z}_{1:T}|\mathbf{z}_0)}\left[\log \frac{p(\mathbf{z}_T)p_\theta(\mathbf{z}_0|\mathbf{z}_1,\mathbf{f})\prod_{t=2}^{T}p_\theta(\mathbf{z}_{t-1}|\mathbf{z}_t,\mathbf{f})}{q(\mathbf{z}_1|\mathbf{z}_0)\prod_{t=2}^{T}q(\mathbf{z}_t|\mathbf{z}_{t-1})}\right] \\
= & \mathbb{E}_{q(\mathbf{z}_{1:T}|\mathbf{z}_0)}\left[\log \frac{p(\mathbf{z}_T)p_\theta(\mathbf{z}_0|\mathbf{z}_1,\mathbf{f})}{q(\mathbf{z}_1|\mathbf{z}_0)} + \log\prod_{t=2}^{T}\frac{p_\theta(\mathbf{z}_{t-1}|\mathbf{z}_t,\mathbf{f})}{q(\mathbf{z}_t|\mathbf{z}_{t-1})}\right] \\
= & \mathbb{E}_{q(\mathbf{z}_{1:T}|\mathbf{z}_0)}\left[\log \frac{p(\mathbf{z}_T)p_\theta(\mathbf{z}_0|\mathbf{z}_1,\mathbf{f})}{q(\mathbf{z}_1|\mathbf{z}_0)} + \log\prod_{t=2}^{T}\frac{p_\theta(\mathbf{z}_{t-1}|\mathbf{z}_t,\mathbf{f})}{\frac{q(\mathbf{z}_{t-1}|\mathbf{z}_t,\mathbf{z}_0)q(\mathbf{z}_t|\mathbf{z}_0)}{q(\mathbf{z}_{t-1}|\mathbf{z}_0)}}\right] \\
= & \mathbb{E}_{q(\mathbf{z}_{1:T}|\mathbf{z}_0)}\left[\log \frac{p(\mathbf{z}_T)p_\theta(\mathbf{z}_0|\mathbf{z}_1,\mathbf{f})}{q(\mathbf{z}_T|\mathbf{z}_0)} + \sum_{t=2}^{T}\log\frac{p_\theta(\mathbf{z}_{t-1}|\mathbf{z}_t,\mathbf{f})}{q(\mathbf{z}_{t-1}|\mathbf{z}_t,\mathbf{z}_0)}\right] \\
= & \mathbb{E}_{q(\mathbf{z}_1|\mathbf{z}_0)}[p_\theta(\mathbf{z}_0|\mathbf{z}_1,\mathbf{f})] + \mathbb{E}_{q(\mathbf{z}_T|\mathbf{z}_0)}\left[\log\frac{p(\mathbf{z}_T)}{q(\mathbf{z}_T|\mathbf{z}_0)}\right] \\
& + \sum_{t=2}^{T}\mathbb{E}_{q(\mathbf{z}_t,\mathbf{z}_{t-1}|\mathbf{z}_0)}\left[\log\frac{p_\theta(\mathbf{z}_{t-1}|\mathbf{z}_t,\mathbf{f})}{q(\mathbf{z}_{t-1}|\mathbf{z}_t,\mathbf{z}_0)}\right] \\
= & \mathbb{E}_{q(\mathbf{z}_1|\mathbf{z}_0)}[p_\theta(\mathbf{z}_0|\mathbf{z}_1,\mathbf{f})] - \mathrm{KL}[q(\mathbf{z}_T|\mathbf{z}_0)\|p(\mathbf{z}_T)] \\
& - \sum_{t=2}^{T}\mathbb{E}_{q(\mathbf{z}_t|\mathbf{z}_0)}[\mathrm{KL}[q(\mathbf{z}_{t-1}|\mathbf{z}_t,\mathbf{z}_0)\|p_\theta(\mathbf{z}_{t-1}|\mathbf{z}_t,\mathbf{f})]]
\end{aligned}
\tag{34}
$$

where we denote $\mathrm{KL}[q(\mathbf{z}_{t-1}|\mathbf{z}_t,\mathbf{z}_0)\|p_\theta(\mathbf{z}_{t-1}|\mathbf{z}_t,\mathbf{f})]$ as $\mathcal{L}_{\mathrm{PSDM},t-1}$, then we have:

$$
\mathcal{L}_{\mathrm{PSDM},t-1} = \mathbb{E}_{\boldsymbol{\epsilon}\sim\mathcal{N}(0,\mathbf{I})}\left[\frac{\beta_t^2}{2\gamma_t^2(1-\beta_t)(1-\alpha_t^2)}\|\boldsymbol{\epsilon} - \boldsymbol{\epsilon}_\theta(\mathbf{z}_t,\mathbf{f},t)\|_2^2\right]
\tag{35}
$$

which gives us the weights of $w(t)$ for $t = 1,\ldots,T$.

For term $\mathbb{E}_{q(\mathbf{z}_1|\mathbf{z}_0)}[p_\theta(\mathbf{z}_0|\mathbf{z}_1,\mathbf{f})]$, we denote as $\mathcal{L}_{\mathrm{PSDM},0}$. With sampling at the final timestep for different modality features and a normalization constant $Z$, we have:

$$
\mathcal{L}_{\mathrm{PSDM},0} = \mathbb{E}_{\boldsymbol{\epsilon}\sim\mathcal{N}(0,\mathbf{I})}\left[\log Z^{-1} - \frac{1}{2}\|\boldsymbol{\epsilon} - \boldsymbol{\epsilon}_\theta(\mathbf{z},\mathbf{f},1)\|^2\right]
\tag{36}
$$

Since $\mathbf{z}_T \sim \mathcal{N}(0,\mathbf{I})$, we have:

$$
\mathcal{L}_{\mathrm{PSDM},T} = \mathrm{KL}[q(\mathbf{z}_T|\mathbf{z}_0)\|p(\mathbf{z}_T)] = 0
\tag{37}
$$

Therefore, we have:

$$
\mathbb{E}_{p_{\langle\mathbf{x},\mathbf{h}\rangle\in\{\mathcal{G}\}},[\mathbf{f}_\mathbf{x},\mathbf{f}_\mathbf{h}]=\mathcal{E}_\phi(\mathcal{G}^f)}[\log p_\theta(\mathbf{z}|\mathbf{f})] \geq -\sum_{t=2}^{T}\mathcal{L}_{\mathrm{PSDM},t-1} - \mathcal{L}_{\mathrm{PSDM},0} = -\mathcal{L}_{\mathrm{PSDM}}
\tag{38}
$$

$\square$

We then prove Equation 28:

*Proof.* $\mathcal{L}$ is $SE(3)$-**invariance**

Our expected outcome is $\forall\mathbf{R}$, $\mathcal{L}(\mathbf{x},\mathbf{h},\mathbf{x}^f,\mathbf{h}^f) = \mathcal{L}(\mathbf{R}\mathbf{x},\mathbf{h},\mathbf{R}\mathbf{x}^f,\mathbf{h}^f)$, and $\forall\mathbf{R}$, $\mathcal{L}_{\mathrm{EAAE}}(\mathbf{x},\mathbf{h},\mathbf{x}^f,\mathbf{h}^f) = \mathcal{L}_{\mathrm{EAAE}}(\mathbf{R}\mathbf{x},\mathbf{h},\mathbf{R}\mathbf{x}^f,\mathbf{h}^f)$ is ensured in Proof. C. For $\mathcal{L}_{\mathrm{PSDM}}$, we expect $\forall\mathbf{R}$, $\mathcal{L}_{\mathrm{PSDM}}(\mathbf{R}\mathbf{z}_{\mathbf{x},0},\mathbf{z}_{\mathbf{h},0},\mathbf{R}\mathbf{f}) = \mathcal{L}_{\mathrm{PSDM}}(\mathbf{z}_{\mathbf{x},0},\mathbf{z}_{\mathbf{h},0},\mathbf{f})$ we have:

$$
\begin{aligned}
& \mathcal{L}_{\mathrm{PSDM}}(\mathbf{R}\mathbf{z}_{\mathbf{x},0},\mathbf{z}_{\mathbf{h},0}) \\
= & \mathbb{E}_{\mathcal{G},\mathcal{E}_\phi}\left[\sum_{t=2}^{T}\mathbb{E}_{q(\mathbf{z}_t|\mathbf{R}\mathbf{z}_0)}[\mathrm{KL}[q(\mathbf{z}_{t-1}|\mathbf{z}_t,\mathbf{R}\mathbf{z}_0)\|p_\theta(\mathbf{z}_{t-1}|\mathbf{z}_t,\mathbf{R}\mathbf{f})]] - \mathbb{E}_{q(\mathbf{z}_1|\mathbf{R}\mathbf{z}_0)}[p_\theta(\mathbf{R}\mathbf{z}_0|\mathbf{z}_1,\mathbf{R}\mathbf{f})]\right]
\end{aligned}
$$

$$= \int_{\mathcal{G}} \left[ \sum_{t=2}^{T} \log \frac{q(\mathbf{z}_{t-1}|q(\mathbf{z}_t, \mathbf{R}\mathbf{z}_0)}{p_\theta(\mathbf{z}_{t-1}|\mathbf{z}_t, \mathbf{R}\mathbf{f})} - \log p_\theta(\mathbf{R}\mathbf{z}_0|\mathbf{z}_1, \mathbf{R}\mathbf{f}) \right]$$

$$= \int_{\mathcal{G}} \left[ \sum_{t=2}^{T} \log \frac{q(\mathbf{R}\mathbf{R}^{-1}\mathbf{z}_{t-1}|q(\mathbf{R}\mathbf{R}^{-1}\mathbf{z}_t, \mathbf{R}\mathbf{z}_0)}{\mathbf{R}\mathbf{R}^{-1}p_\theta(\mathbf{z}_{t-1}|\mathbf{R}\mathbf{R}^{-1}\mathbf{z}_t, \mathbf{R}\mathbf{f})} - \log p_\theta(\mathbf{R}\mathbf{z}_0|\mathbf{R}\mathbf{R}^{-1}\mathbf{z}_1, \mathbf{R}\mathbf{f}) \right] \quad \mathbf{R}\mathbf{R}^{-1} = \mathbf{I}$$

$$= \int_{\mathcal{G}} \left[ \sum_{t=2}^{T} \log \frac{q(\mathbf{R}^{-1}\mathbf{z}_{t-1}|q(\mathbf{R}^{-1}\mathbf{z}_t, \mathbf{z}_0)}{\mathbf{R}^{-1}p_\theta(\mathbf{z}_{t-1}|\mathbf{R}^{-1}\mathbf{z}_t, \mathbf{f})} - \log p_\theta(\mathbf{z}_0|\mathbf{R}^{-1}\mathbf{z}_1, \mathbf{f}) \right] \quad SE(3) \text{ of } \mathbf{f}_\mathbf{x} \, \& \, \mathbf{z}_t$$

$$= \mathbb{E}_{\mathcal{G}, \mathcal{E}_\phi} \left[ \sum_{t=2}^{T} \log \frac{q(\mathbf{j}_{t-1}|q(\mathbf{j}_t, \mathbf{z}_0)}{\mathbf{R}^{-1}p_\theta(\mathbf{z}_{t-1}|\mathbf{j}_t, \mathbf{f})} - \log p_\theta(\mathbf{z}_0|\mathbf{j}_1, \mathbf{f}) \right] \quad \text{Let } \mathbf{j}_t = \mathbf{R}^{-1}\mathbf{z}_t$$

$$= \mathcal{L}_{\text{PSDM}}(\mathbf{z}_{\mathbf{x},0}, \mathbf{z}_{\mathbf{h},0}, \mathbf{f})$$

$$\tag{39}$$

$$\square$$

## E   TRAINING DETAILS

**Parameters**

1. Optimizer: Adam (Kingma & Ba, 2015) optimizer is used with a constant learning rate of $10^{-4}$ as our default training configuration.
2. Batch size: 64.
3. EGNN in PSDM: 9 layers and 256 hidden features for QM9, 4 layers and 256 hidden features for GEOM-DRUG.
4. EGNN in EAAE: 1 layer and 256 hidden features for the encoder for QM9 and GEOM-DRUG, 9 layers and 4 layers with 256 hidden features for the decoder for QM9 and GEOM-DRUG, respectively.
5. Latent dimension of $\mathbf{f}_\mathbf{x}, \mathbf{f}_\mathbf{h}$: latent dimension is 3 and 1 for $\mathbf{f}_\mathbf{x}$ and $\mathbf{f}_\mathbf{h}$, respectively.
6. Epoch: 3000 for QM9 and 10 for GEOM-DRUG.

**Training**

1. GPU: NVIDIA GeForce RTX 3090
2. CPU: Intel(R) Xeon(R) Platinum 8338C CPU
3. Memory: 512 GB
4. Time: Around 7 days for QM9 and 20 days for GEOM-DRUG.

**Specific Parameters** 1. On QM9, we train PSDM with 9 layers and 256 hidden features with a batch size 64; 2. On GEOM-DRUG, we train PSDM with 4 layers and 256 hidden features, with batch size 64;

## F   ALGORITHMS

This section contains two main algorithms of the proposed *GODD*. Algorithm 1 presents the pseudo-code for training *GODD* on the in distributional training data set $\{\mathcal{G}_I\}$ and corresponding fragment set $\{\mathcal{G}_I^f\}$. Algorithm 2 presents the process of OOD molecule generation using the ODD scaffold/ring $\mathcal{G}_O^f$.

## G   *QM9* DATASET

QM9 (Ramakrishnan et al., 2014) is a comprehensive dataset that provides geometric, energetic, electronic, and thermodynamic properties for a subset of the GDB-17 database (Ruddigkeit et al., 2012), comprising 134 thousand stable organic molecules with up to nine heavy atoms.

---

**Algorithm 1** Training *GODD*

---

1: **Input:** in-distribution geometric data point $\mathcal{G}_I = \langle \mathbf{x}, \mathbf{h} \rangle$, corresponding fragment $\mathcal{G}_I^f$, asymmetric encoder $\mathcal{E}_\phi$ and decoder $\mathcal{D}_\vartheta$, denoising network $\epsilon_\theta$;
2: **EAAE:**
3: $\boldsymbol{\mu_x}, \mu_\mathbf{h} \leftarrow \mathcal{E}_\phi(\mathbf{x}^f, \mathbf{h}^f)$      *// Encode*
4: $\langle \boldsymbol{\epsilon_x}, \boldsymbol{\epsilon_h} \rangle \sim \mathcal{N}(\mathbf{0}, \mathbf{I})$      *// Sample Noise for EAAE*
5: $\boldsymbol{\epsilon_x} \leftarrow \boldsymbol{\epsilon_x} - \mathbf{G}(\boldsymbol{\epsilon_x})$      *// Subtract Center of Gravity*
6: $\mathbf{f_x}, \mathbf{f_h} \leftarrow \mu + \langle \boldsymbol{\epsilon_x}, \boldsymbol{\epsilon_h} \rangle \odot \sigma_0$      *// Reparameterization*
7: **PSDM:**
8: $t \sim \mathcal{U}(0, T)$      *// Sample Timestep*
9: $\langle \boldsymbol{\epsilon_x}, \boldsymbol{\epsilon_h} \rangle \sim \mathcal{N}(\mathbf{0}, \mathbf{I})$      *// Sample Noise for PSDM*
10: $\boldsymbol{\epsilon_x} \leftarrow \boldsymbol{\epsilon_x} - \mathbf{G}(\boldsymbol{\epsilon_x})$      *// Subtract Center of Gravity*
11: $\mathbf{z}_{\mathbf{x},t}, \mathbf{z}_{\mathbf{h},t} \leftarrow \alpha_t[\mathbf{x}, \mathbf{h}] + \sigma_t \boldsymbol{\epsilon}$      *// Diffuse*
12: $\hat{\mathbf{x}}, \hat{\mathbf{h}} \leftarrow \mathcal{D}_\vartheta(\mathbf{f_x}, \mathbf{f_h})$      *// Decode*
13: **Optimization**
14: $\mathcal{L}_{\text{EAAE}} \leftarrow \mathcal{L}([\hat{\mathbf{x}}, \hat{\mathbf{h}}], [\mathbf{x}, \mathbf{h}]) + \text{KL}$      *// $\mathcal{L}$ for EAAE*
15: $\mathcal{L}_{\text{PSDM}} \leftarrow \| \boldsymbol{\epsilon} - \boldsymbol{\epsilon}_\theta(\mathbf{z}_{\mathbf{x},t}, \mathbf{z}_{\mathbf{h},t}, t, \mathbf{f_x}, \mathbf{f_h}) \|^2$      *// $\mathcal{L}$ for PSDM*
16: $\mathcal{L}_{GODD} \leftarrow \mathcal{L}_{\text{EAAE}} + \mathcal{L}_{\text{PSDM}}$      *// Total Loss*
17: $\phi, \vartheta, \theta \leftarrow \text{optimizer}(\mathcal{L}_{GODD}, \phi, \vartheta, \theta)$      *// Optimize*
18: **return** $\phi, \theta$

---

**Algorithm 2** Adaptive Sampling of *GODD*

---

1: **Input:** OOD fragment $\mathcal{G}_O^f = \langle \mathbf{x}_O^f, \mathbf{h}_O^f \rangle$, encoder $\mathcal{E}_\phi$, denoising network $\epsilon_\theta$;
2: $\boldsymbol{\mu_x}, \mu_\mathbf{h} \leftarrow \mathcal{E}_\phi(\mathbf{x}_O^f, \mathbf{h}_O^f)$      *// Encode*
3: $\langle \boldsymbol{\epsilon_x}, \boldsymbol{\epsilon_h} \rangle \sim \mathcal{N}(\mathbf{0}, \mathbf{I})$      *// Sample Noise for EAAE*
4: $\boldsymbol{\epsilon_x} \leftarrow \boldsymbol{\epsilon_x} - \mathbf{G}(\boldsymbol{\epsilon_x})$      *// Subtract Center of Gravity*
5: $\mathbf{f_x}, \mathbf{f_h} \leftarrow \mu + \langle \boldsymbol{\epsilon_x}, \boldsymbol{\epsilon_h} \rangle \odot \sigma_0$      *// Target Condition*
6: $\langle \mathbf{z}_{\mathbf{x},T}, \mathbf{z}_{\mathbf{h},T} \rangle \sim \mathcal{N}(\mathbf{0}, \mathbf{I})$      *// Sample Noise for Generation*
7: **for** $t$ **in** $T, T-1, \dots, 1$ **do**
8:     $\langle \boldsymbol{\epsilon_x}, \boldsymbol{\epsilon_h} \rangle \sim \mathcal{N}(\mathbf{0}, \mathbf{I})$      *// Denoising*
9:     $\boldsymbol{\epsilon_x} \leftarrow \boldsymbol{\epsilon_x} - \mathbf{G}(\boldsymbol{\epsilon_x})$      *// Subtract Center of Gravity*
10:     $\mathbf{z}_{\mathbf{x},t-1}, \mathbf{z}_{\mathbf{h},t-1} \leftarrow \frac{1}{\sqrt{1-\beta_t}}(\langle \mathbf{z}_{\mathbf{x},t}, \mathbf{z}_{\mathbf{h},t} \rangle - \frac{\beta_t}{\sqrt{1-\alpha_t^2}} \boldsymbol{\epsilon}_\theta(\mathbf{z}_{\mathbf{x},t}, \mathbf{z}_{\mathbf{h},t}, t, \mathbf{f_x}, \mathbf{f_h})) + \rho_t \boldsymbol{\epsilon}$
11: **end for**
12: $\mathbf{x}, \mathbf{h} \leftarrow p(\mathbf{z}_{\mathbf{x},0}, \mathbf{z}_{\mathbf{h},0} | \mathbf{z}_{\mathbf{x},1}, \mathbf{z}_{\mathbf{h},1}, \mathbf{f_x}, \mathbf{f_h})$
13: **return** $\langle \mathbf{x}, \mathbf{h} \rangle$

---

### G.1 SCAFFOLD SPLIT QM9

We utilized the open-source software, RDkit (Landrum et al., 2016), to examine the scaffold and ring of each molecule. QM9 dataset [1] comprises a total of 130,831 molecules, encompassing 15,661 unique scaffolds. Molecules lacking a scaffold were denoted as '-' and included in the total scaffold count. The dataset was divided based on scaffold frequency. Specifically, the in-distribution subset contained 100,000 molecules and 1,054 scaffolds. The OOD I subset included 15,000 molecules and 2,532 scaffolds, while the OOD II subset consisted of 15,831 molecules and 12,075 scaffolds.

Figure 4(a) presents the division's schematic diagram. Figure 4(b) displays the logarithmic histogram of the scaffolds in each dataset segment. It is evident that the in-distribution dataset contains the most frequent scaffolds, primarily concentrated above 100. The frequency of scaffolds in the OOD I dataset ranges between 10 and 100. In contrast, the scaffolds in the OOD II dataset are primarily concentrated within 10, with most appearing only once. Figures, SMILES, and frequencies of some example scaffolds in each sub-dataset are given in Figure 5.

---

[1]https://springernature.figshare.com/ndownloader/files/3195389

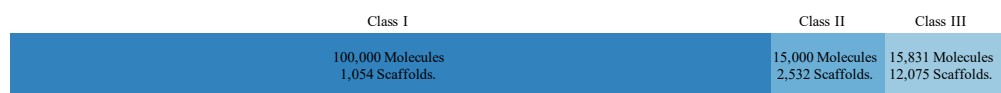

(a) The number of molecules and scaffolds in distribution, OOD I, and OOD II of the Scaffold-Split QM9 data set.

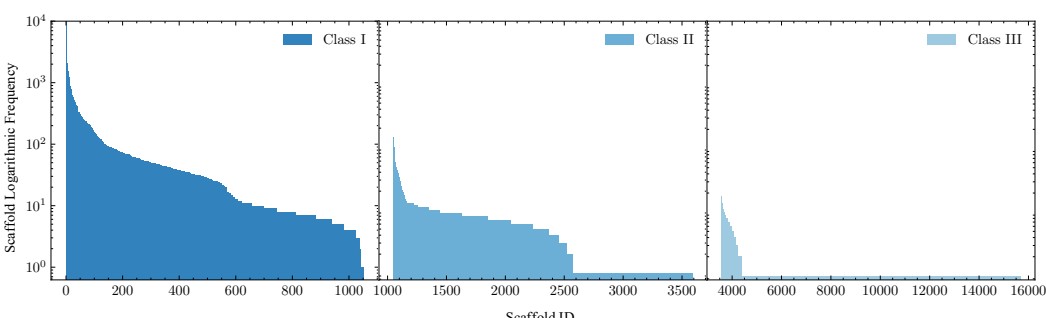

(b) Scaffold Logarithmic Histogram of Scaffold-Split QM9

Figure 4: Scaffold-Split QM9

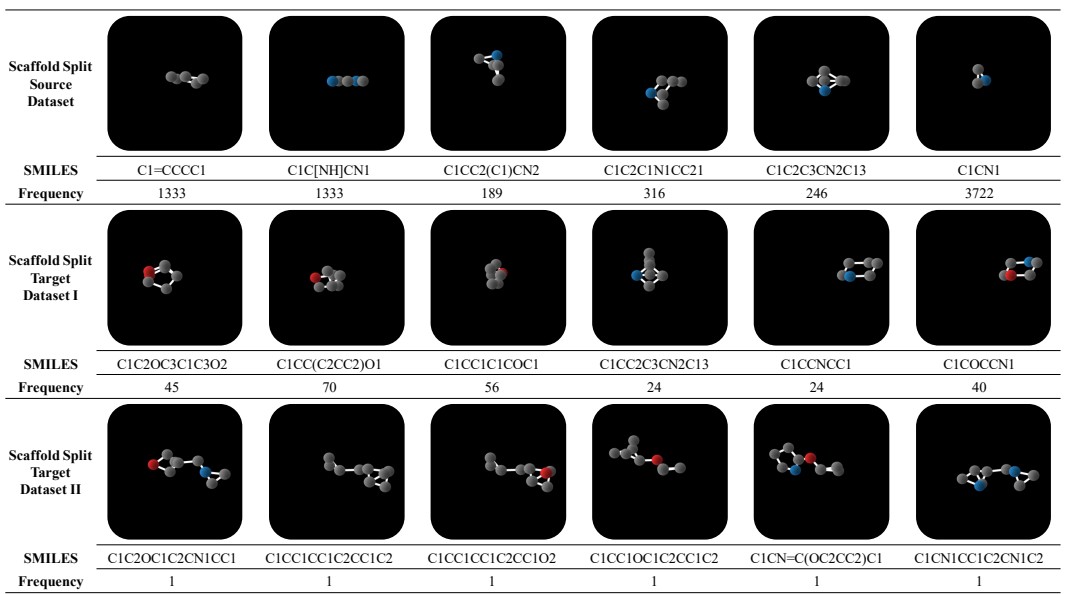

Figure 5: Scaffold Examples of QM9 Split by Scaffolds.

### G.2 RING NUMBER SPLIT QM9

The QM9 dataset could categorize molecules into nine groups based on the number of rings, ranging from 0 to 8. As the number of rings increases, the quantity of molecules correspondingly decreases. We partition the QM9 dataset into two subsets based on ring count. The in-distribution dataset comprises acyclic molecules and those with 1 to 3 rings, while the OOD dataset includes molecules with 4 to 8 rings. Figure 6 presents a schematic diagram illustrating example molecules with 0 to 8 rings.

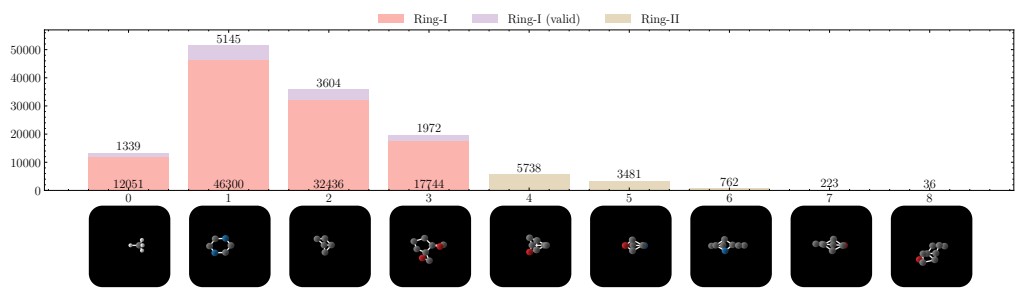

Figure 6: Ring Examples of QM9 Split by Ring Number.

## H  *GEOM-DRUG* DATASET

*GEOM-DRUG* (Geometric Ensemble Of Molecules) dataset (Axelrod & Gómez-Bombarelli, 2022) encompasses around 450,000 molecules, each with an average of 44.2 atoms and a maximum of 181 atoms[2].

### H.1  RING NUMBER SPLIT GEOM-DRUG

The GEOM-DRUG dataset classifies molecules into sixteen categories based on the number of rings, ranging from 0 to 14 and 22. As the ring count increases, the number of molecules correspondingly decreases. The dataset is partitioned into two subsets according to ring count: the in-distributional dataset, which includes molecules with 0 to 10 rings and a count exceeding 100, and four OOD datasets, which comprises molecules with 11 to 14 and 22 rings. Figure 7 provides a schematic representation of the molecule distribution by ring number.

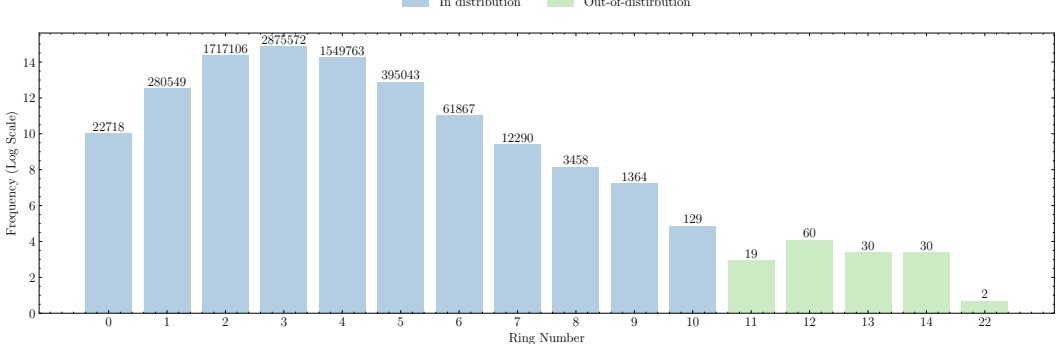

Figure 7: Ring Distribution of GEOM-DRUG dataset.

## I  *GEOM-LINKER* DATASET

The GEOM-LINKER dataset for linker design is constructed by (Igashov et al., 2024) based on GEOM-DRUG. The authors decomposed the molecule into three or more fragments with one or two linkers connecting them. The dataset contains 41,907 molecules and 285,140 fragments, and the original dataset is randomly split into train (282,602 examples), validation (1,250 examples), and test (1,288 examples) sets. Atom types considered for this dataset are C, O, N, F, S, Cl, Br, I, and P.

We present the distribution of molecules in GEOM-LINKER according to the number of rings in Figure 8. The diagram illustrates the molecules with 3 to 5 rings are the majority and molecules

---

[2]https://dataverse.harvard.edu/file.xhtml?fileId=4360331&version=2.0

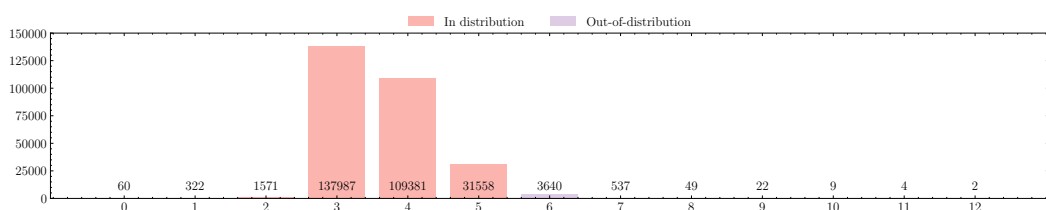

Figure 8: Ring Distribution of GEOM-LINKER dataset.

with 8 to 12 rings exhibit data sparsity in the whole dataset. Thereby, we split the dataset according to the ring numbers into in-distribution (0-5 rings, 280,879 samples) and OOD (6-12 rings, 4,263 samples).

## J    FULL RESULTS OF OOD RING-STRUCTURE MOLECULE GENERATION

We present the detailed quantitative evaluation results of ring adaptive molecule generation tasks in Tables 8 and 9. The results show that the proposed method has dominant performance in all metrics, including ring number proportion, validity, novelty, and success rate.

It is significant to note that the entire QM9 dataset comprises only 36 eight-ring molecules. When the proposed algorithm utilizes the ring structures of these 36 8-ring molecules as input, the target validity reaches an impressive 72.2%, and the novelty is as high as 80.9%. Considering that there are only 36 fundamental 8-ring structures, the uniqueness is slightly lower (27.4%). Nevertheless, the generation of 10,000 molecules resulted in 2,388 valid, unique, and entirely novel eight-ring molecules, which is a substantial breakthrough compared to existing methods (even those models trained on eight-ring molecules) that failed to discover any new eight-ring molecules.

Table 8: Results of molecule proportion in terms of ring-number (P) and molecule validity (V) The **best** results are highlighted in bold. QM9 only contains 36 eight-ring molecules and the proportion for eight-ring is nearly 0.

| | 0 | 1 | 2 | 3 | 4 | 5 | 6 | 7 | 8 | Averaged |
|---|---|---|---|---|---|---|---|---|---|---|
| Method | | | | | P (%) | | | | | - |
| QM9 | 10.2 | 39.3 | 27.6 | 15.1 | 4.4 | 2.7 | 0.6 | 0.2 | 0.0 | - |
| EDM† | 10.5 | 39.8 | 28.0 | 14.5 | 4.0 | 2.9 | 0.2 | 0.1 | 0.0 | - |
| GeoLDM† | 12.0 | 38.6 | 27.0 | 15.3 | 4.6 | 2.2 | 0.2 | 0.1 | 0.0 | - |
| EDM‡ | 12.1 | 44.1 | 29.8 | 11.8 | 1.7 | 0.5 | 0.0 | 0.0 | 0.0 | - |
| GeoLDM‡ | 2.8 | 41.5 | 32.1 | 15.7 | 4.7 | 2.7 | 0.3 | 0.1 | 0.0 | - |
| C-EDM‡ | 98.9 | 94.2 | 80.8 | 64.4 | 12.6 | 26.8 | 0.3 | 0.1 | 0.0 | - |
| C-GeoLDM‡ | 97.1 | 89.4 | 74.2 | 52.4 | 22.3 | 22.7 | 0.9 | 0.2 | 0.0 | - |
| EEGSDE‡ | 98.4 | 92.2 | 77.6 | 58.2 | 14.1 | 17.6 | 0.3 | 0.0 | 0.0 | - |
| MOOD‡ | 80.7 | 87.1 | 86.1 | 73.3 | 34.1 | 32.3 | 10.3 | 0.2 | 0.0 | - |
| CGD‡ | 82.3 | 84.8 | 86.2 | 83.6 | 34.4 | 22.4 | 10.3 | 10.1 | 0.0 | - |
| *GODD*‡ | **99.9** | **99.8** | **99.1** | **97.6** | **92.5** | **89.7** | **78.7** | **88.2** | **82.1** | - |
| | | | | | Target Valid (%) | | | | | |
| QM9 | 97.7 | 97.7 | 97.7 | 97.7 | 97.7 | 97.7 | 97.7 | 97.7 | 97.7 | 97.7 |
| EDM† | 10.8 | 36.1 | 26.7 | 13.9 | 4.0 | 2.3 | 0.2 | 0.1 | 0.0 | 10.5 |
| GeoLDM† | 11.2 | 36.2 | 25.2 | 14.3 | 4.3 | 2.0 | 0.2 | 0.1 | 0.0 | 10.4 |
| EDM‡ | 11.4 | 41.4 | 28.0 | 11.1 | 1.6 | 0.5 | 0.0 | 0.0 | 0.0 | 10.4 |
| GeoLDM‡ | 2.7 | 38.8 | 30.0 | 14.7 | 4.4 | 2.6 | 0.3 | 0.1 | 0.0 | 10.4 |
| C-EDM‡ | 86.6 | 85.4 | 74.9 | 59.8 | 12.1 | 23.3 | 0.2 | 0.1 | 0.0 | 38.0 |
| C-GeoLDM‡ | 86.2 | 79.6 | 65.8 | 48.1 | 20.4 | 20.7 | 0.9 | 0.2 | 0.0 | 35.7 |
| EEGSDE‡ | 96.7 | 92.1 | 77.2 | 58.0 | 13.9 | 17.4 | 0.3 | 0.0 | 0.0 | 39.5 |
| MOOD‡ | 75.5 | 81.7 | 80.6 | 68.9 | 32.0 | 30.1 | 9.6 | 0.1 | 0.0 | 42.1 |
| CGD‡ | 77.0 | 79.6 | 81.1 | 78.4 | 32.3 | 20.9 | 9.5 | 9.5 | 0.0 | 43.2 |
| *GODD*‡ | **31.7** | **91.4** | **91.4** | **92.1** | **85.3** | **85.2** | **69.5** | **82.5** | **72.2** | 77.9 |

Table 9: Results of molecule proportion in terms of novelty (N) and success rate (S). The **best** results are highlighted in bold.

| | 0 | 1 | 2 | 3 | 4 | 5 | 6 | 7 | 8 | Averaged |
|---|---|---|---|---|---|---|---|---|---|---|
| Method | | | | | Target Novelty (%) | | | | | |
| EDM† | 7.1 | 23.6 | 17.5 | 9.1 | 2.6 | 1.5 | 0.1 | 0.1 | 0.0 | 6.8 |
| GeoLDM† | 7.0 | 22.4 | 15.6 | 8.9 | 2.7 | 1.3 | 0.1 | 0.0 | 0.0 | 6.4 |
| EDM‡ | 7.5 | 27.1 | 18.3 | 7.2 | 1.1 | 0.3 | 0.0 | 0.0 | 0.0 | 6.8 |
| GeoLDM‡ | 1.7 | 25.0 | 19.4 | 9.5 | 2.8 | 1.7 | 0.2 | 0.1 | 0.0 | 6.7 |
| C-EDM‡ | 57.1 | 59.7 | 54.2 | 44.2 | 9.9 | 20.1 | 0.2 | 0.1 | 0.0 | 27.3 |
| C-GeoLDM‡ | 63.3 | 61.6 | 53.3 | 40.1 | 17.3 | 18.3 | 0.7 | 0.1 | 0.0 | 28.3 |
| EEGSDE‡ | 63.9 | 61.4 | 53.0 | 42.5 | 9.9 | 14.1 | 0.3 | 0.0 | 0.0 | 27.2 |
| MOOD‡ | 50.0 | 53.9 | 53.6 | 44.4 | 20.6 | 20.0 | 6.3 | 0.1 | 0.0 | 27.6 |
| CGD‡ | 51.0 | 52.5 | 53.1 | 51.3 | 21.0 | 13.9 | 6.3 | 6.2 | 0.0 | 28.4 |
| *GODD*‡ | **96.6** | **51.3** | **55.6** | **60.2** | **69.5** | **63.5** | **71.5** | **83.4** | **80.9** | **70.3** |
| | | | | | Success Rate (%) | | | | | |
| EDM† | 6.5 | 21.9 | 16.2 | 8.4 | 2.4 | 1.4 | 0.1 | 0.1 | 0.0 | 6.3 |
| GeoLDM† | 6.4 | 20.6 | 14.4 | 8.2 | 2.4 | 1.2 | 0.1 | 0.0 | 0.0 | 5.9 |
| EDM‡ | 6.9 | 25.1 | 17.0 | 6.7 | 1.0 | 0.3 | 0.0 | 0.0 | 0.0 | 6.3 |
| GeoLDM‡ | 1.6 | 23.0 | 17.8 | 8.7 | 2.6 | 1.5 | 0.2 | 0.1 | 0.0 | 6.1 |
| C-EDM‡ | 48.1 | 53.8 | 50.0 | 40.5 | 7.9 | 16.8 | 0.2 | 0.1 | 0.0 | 24.1 |
| C-GeoLDM‡ | 54.6 | 54.6 | 46.9 | 36.8 | 15.4 | 15.6 | 0.6 | 0.1 | 0.0 | 25.0 |
| EEGSDE‡ | 54.7 | 54.7 | 46.9 | 39.5 | 9.5 | 12.2 | 0.2 | 0.0 | 0.0 | 24.2 |
| MOOD‡ | 45.9 | 49.8 | 49.4 | 41.0 | 18.9 | 18.3 | 5.8 | 0.1 | 0.0 | 25.5 |
| CGD‡ | 46.8 | 48.5 | 49.1 | 47.3 | 19.5 | 12.8 | 5.8 | 5.7 | 0.0 | 26.2 |
| *GODD*‡ | **25.9** | **43.4** | **46.2** | **50.4** | **53.8** | **41.0** | **46.1** | **34.1** | **23.9** | **40.5** |

## K VISUALIZATION

In this section, we provide additional visualizations of physical prior steered molecule generation by *GODD* for OOD scaffold generation and ring number generation in Figures 9 and 10

As depicted in the two figures, the model consistently generates realistic molecular geometries with OOD scaffolds or ring numbers.

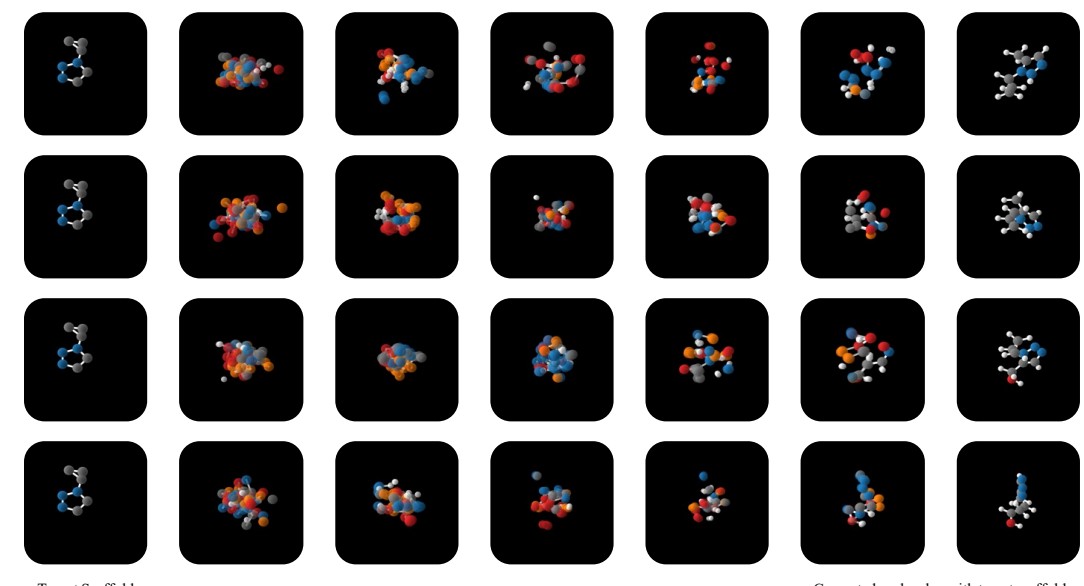

Target Scaffold                                                       Generated molecules with target scaffold

Figure 9: Molecules Generated by *GODD* for Scaffold Adaptive Generation Under The Same Unseen Scaffold Condition.

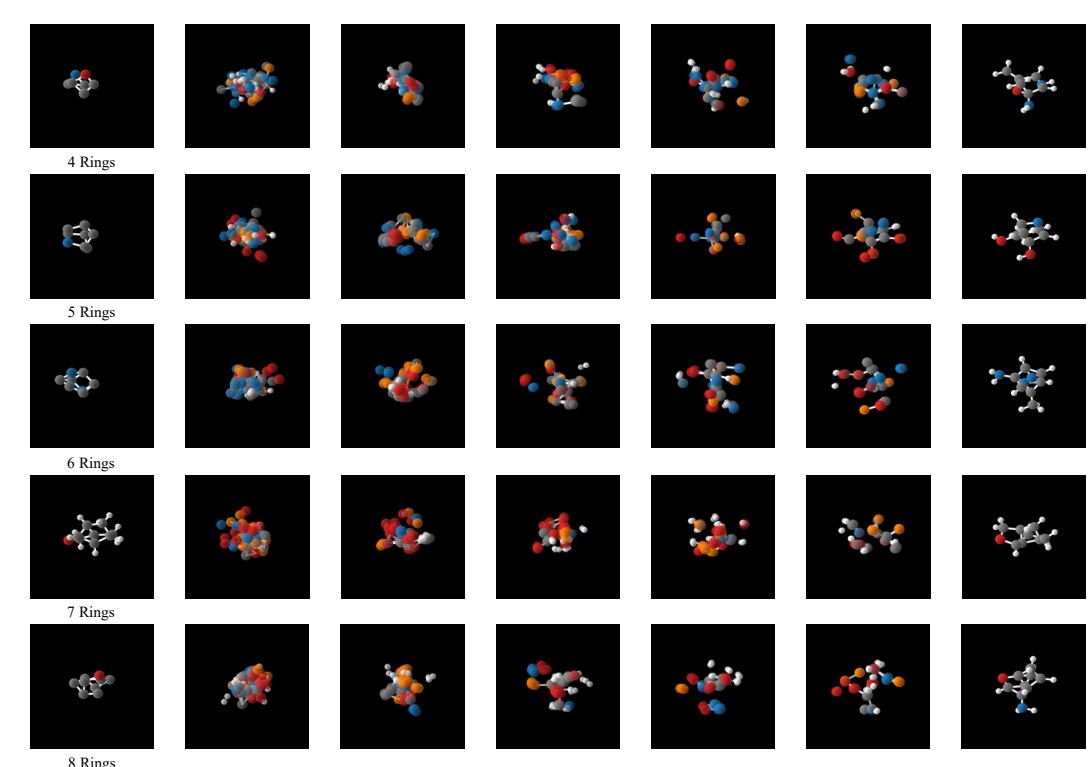

4 Rings

5 Rings

6 Rings

7 Rings

8 Rings

Figure 10: Molecules Generated by *GODD* for Ring Number Adaptive Generation For Unseen Ring Numbers

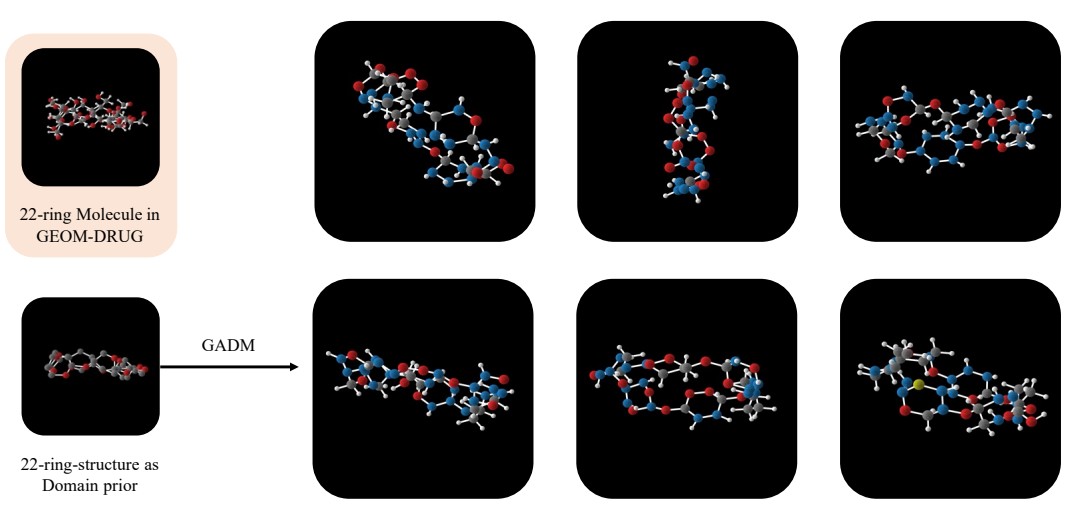

22-ring Molecule in GEOM-DRUG

22-ring-structure as Domain prior

GADM

Figure 11: Molecules Generated by *GODD* for Ring Number Adaptive Generation For Unseen Ring Numbers on GEOM-DRUG Dataset.

## L  RELATED WORK

**Molecule Generation Models.**  Prior studies on molecule generation focused on generating molecules as 2D graphs (Jin et al., 2018; Liu et al., 2018; Shi et al., 2020). However, there has been a growing interest in 3D molecule generation. G-SchNet (Gebauer et al., 2019) and G-SphereNet (Luo & Ji, 2022) utilize autoregressive techniques to construct molecules incrementally by progressively connecting atoms or molecular fragments. These frameworks necessitate either a meticulous formulation of complex action space or action ordering.

More recently, the focus has shifted towards using Diffusion Models (DMs) for 3D molecule generation (Hoogeboom et al., 2022; Xu et al., 2023; Wu et al., 2022; Song et al., 2024). To mitigate the inconsistency of unified Gaussian diffusion across diverse modalities, a latent space was introduced by (Xu et al., 2023). To tackle the atom-bond inconsistency problem, different noise schedulers were proposed by (Peng et al., 2023) for various modalities to accommodate noise sensitivity. However, these algorithms do not account for generating novel molecules outside the training distribution.

**Out-of-Distribution Molecule Generation.**  OOD generation, although under-explored, is of paramount importance, especially considering that molecules generated by machine-learning methods often exhibit a "striking similarity" (Walters & Murcko, 2020). In recent years, some preliminary work has begun to use reinforcement learning (Yang et al., 2021) and out-of-distribution control (Lee et al., 2023) to explore the generation of novel molecules. However, these methods are still challenging when designing novel molecules in data-sparse regions with fragment shifts. As proposed by (Lee et al., 2023), MOOD employs an OOD control and integrates a property-predictor-based diffusion scheme to optimize molecules for specific chemical properties. Similarly, CGD (Klarner et al., 2024) leverages unlabeled data to improve the generalization of guided diffusion models. However, these predictor-based OOD methods fail to generate novel molecules with ODD fragments that are sparse for training a classifier.

**Fragment-Based Drug Design.**  The discovery of new molecules is crucial across various fields, and there are four primary approaches to this task (Murray & Rees, 2009): (1) searching from an existing molecule, (2) developing from a natural product, (3) high-throughput screening, and (4) fragment-based drug discovery (FBDD). Among these, FBDD has gained significant importance and interest over the past decades due to its higher efficiency compared to other methods (Murray & Rees, 2009). Typically, fragments are selected based on the "rule of three" (Congreve et al., 2003) criteria and thereby can be grown, linked, or merged to develop potential molecules (Bian & Xie, 2018). Recently, there has been a growing trend in enhancing FBDD with machine learning techniques (Wu et al., 2024; Igashov et al., 2024; Guan et al., 2024). However, these methods often overlook the issue of fragment sparsity in datasets, highlighting the need for an OOD molecular generative model capable of producing realistic molecules in data-sparse regions.

## M  IMPACT STATEMENTS

This paper presents work whose goal is to advance the field of generative Artificial Intelligence (AI) for scientific fields, such as material science, chemistry, and biology. The obtained experience/knowledge will greatly boost generative AI technologies in facilitating the process of scientific knowledge discovery.

Machine learning for molecule generation opens up possibilities for designing molecules beyond therapeutic purposes, such as the creation of illicit drugs or dangerous substances. The potential for misuse and unintended consequences necessitates strict ethical guidelines, robust regulation, and responsible use of these technologies to prevent harm to individuals and society.

## N  ACRONYMS LIST

### ACRONYMS

***GODD***  Geometric OOD Diffusion Model. 1–10, 18, 20, 21, 25–27

**EAAE**  Equivariant Asymmetric Autoencoder. 4–6, 9, 15, 17–21

**PSDM**  Physical Prior Steered Diffusion Model. 6, 18–21

