# OpenReview forum: "Steering 3D Molecule Generation in Data-Sparse Regions via Distributional Physical Priors"
_ICLR.cc/2025/Conference — Submitted to ICLR 2025_

### Official Review · Reviewer_kCta · 2024-10-24

**Soundness:** 3
**Presentation:** 2
**Contribution:** 3
**Rating:** 6
**Confidence:** 3

**Summary:**

The paper proposes a framework capable of generating molecules in out-of-domain distributions, specifically, those OOD distributions driven by scaffold and ring structures. The framework utilizes an asymmetric autoencoder to capture distributional physical priors (specifically, point-wise latent representations) and employs these priors to guide downstream generative models. The authors demonstrate the effectiveness of their approach through extensive experiments on the QM9, GEOM-DRUG and GEOM-LINK datasets.

**Strengths:**

1. Tackling the task of generating OOD molecules, specifically focusing on scaffolds not seen during training, is a valuable and practically important challenge, especially in applications like drug design. This work makes significant progress in formulating and addressing this problem.
2. The experimental performance of the proposed GODD framework shows promising results.
3. The codes that the authors provide seem well organized and ready-to-run (though I did not try to reproduce it).

**Weaknesses:**

1. Ambiguous Statements: The definitions of $p_\theta(z_x, z_h, f_x, f_h)$ and $\epsilon_\theta(z_x, z_h, t, f_x, f_h)$ are unclear. Specifically, how are $f_x$ and $f_h$ integrated into the noisy point clouds $z_x$ and $z_h$, and how does the EGNN ($\epsilon_\theta$) operate on this mixed point cloud? This is a critical aspect of the method but is not discussed in sufficient detail (correct me if I missed something). Additionally, Figure 1 is somewhat confusing: It depicts the physical prior as a global-level representation, whereas it is actually a latent point cloud, which is conceptually quite different (lots of previous and concurrent work has managed the first case).
2. Potentially Unfair Comparisons: The experimental comparisons may be somewhat skewed. For example, in Tables 2 and 3, models like EDM, GeoLDM, and EquiFM condition on nothing, so it’s not surprising that the percentage (P%) of OOD generation is small, which likely reflects the rarity of OOD structures in the overall distribution. This is even pronounced when these models weren’t even trained on the entire dataset. While the authors attempt to construct comparable baselines such as C-EDM, conditioning on a single number seems less fair compared to the geometry elements (referred to as physical priors) that GODD conditions on. In fact, these geometry elements are the targets we aim to generate, so evaluating the partition of these elements in the output when they are taken as input may not be as fair.

**Questions:**

See “Weaknesses”.

---

> ### Author Response · Authors · 2024-11-24
> **Response to Reviewer kCta (1/2)**
>
> We appreciate the reviewer’s acknowledgment of the strengths of our work. We are glad that the reviewer recognizes **the value and practical importance of our research**, **finds our results promising**, and **commends the organization of our code**. Below, we address the reviewer’s concerns.
>
> # Response to Weakness 1
> > Ambiguous Statements: The definitions of $p_{\\theta}(\\mathbf{z}\_{\\mathbf{x}},\\mathbf{z}\_{\\mathbf{h}},\\mathbf{f}\_{x}, \\mathbf{f}\_{\\mathbf{h}})$ and $\\epsilon\_{\\theta}(\\mathbf{z}\_{\\mathbf{x}},\\mathbf{z}\_{\\mathbf{h}},\\mathbf{f}\_{\\mathbf{x}}, \\mathbf{f}\_{\\mathbf{h}})$ are unclear ...
>
> We appreciate the reviewer’s comments.
>
> 1. The term $p\_{\\theta}(\\mathbf{z}\_{\\mathbf{x},t-1}, \\mathbf{z}\_{\\mathbf{h},t-1}|\\mathbf{z}\_{\\mathbf{x},t}, \\mathbf{z}\_{\\mathbf{h},t},\\mathbf{f}\_{\\mathbf{x}},\\mathbf{f}\_{\\mathbf{h}})$ represents the conditional reverse process of our diffusion model, analogous to $p\_{\\theta}(x\_{t-1}|x\_{t})$ in DDPM. The key distinction is that our model is additionally conditioned on encoded physical priors $\\mathbf{f}\_{\\mathbf{x}}$ and $\\mathbf{f}\_{\\mathbf{h}}$. Besides, $\\boldsymbol{\\epsilon}\_{\\theta}(\\mathbf{z}\_{\\mathbf{x},t}, \\mathbf{z}\_{\\mathbf{h},t},\\mathbf{f}\_{\\mathbf{x}},\\mathbf{f}\_{\\mathbf{h}},t)$ denotes the denoising network, which incorporates these physical priors as extra inputs.
>
> 2. Given fragment inputs $\\mathcal{G}^{f}=\\langle\\mathbf{x}^{f},\\mathbf{h}^{f}\\rangle$ and the corresponding learned physical prior $\\mathcal{F}=\\langle\\mathbf{f}\_{\\mathbf{x}},\\mathbf{f}\_{\\mathbf{h}}\\rangle$, we concatenate $\\mathbf{f}\_{\\mathbf{x}}$ and $\\mathbf{f}\_{\\mathbf{h}}$ to the node features $\\mathbf{z}\_{h}$. For instance, with node features $\\mathbf{z}\_{h} \\in \\mathbb{R}^{N \\times d}$, $\\mathbf{f}\_{\\mathbf{x}} \\in \\mathbb{R}^{N' \\times 3}$, and $\\mathbf{f}\_{\\mathbf{h}} \\in \\mathbb{R}^{N' \\times 1}$, the EGNN within the denoising network $\\boldsymbol{\\epsilon}\_{\\theta}$ receives noisy coordinates $\\mathbf{z}\_{x} \\in \\mathbb{R}^{N \\times 3}$ and concatenated features $\\mathbf{z}\_{h} \\in \\mathbb{R}^{N \\times (d+3+1+1)}$ (including a extra one-dimensional time-step embedding). Since the number of fragments $N'$ is smaller than the number of molecules $N$, zeros are padded to $\\mathbf{f}\_{\\mathbf{x}}$ and $\\mathbf{f}\_{\\mathbf{h}}$. We supplemented this explanation to the revised manuscript on Page 6.
>
> 3. Regarding Figure 1, we agree that the physical prior is indeed a latent point cloud. However, in our specific implementation, this latent point cloud is concatenated into the node features to guide the generation process. Thus, we chose to depict it as a representation to effectively convey our concept.

---

> ### Author Response · Authors · 2024-11-24
> **Response to Reviewer kCta (2/2)**
>
> # Response to Weakness 2
> > W2. Potentially Unfair Comparisons: The experimental comparisons may be somewhat skewed ...
>
> We appreciate the reviewer’s comments and recognize the inherent challenges in achieving completely fair comparisons in OOD settings, especially as we are pioneering this new area of research.
>
> First, prior works can only realize conditional generation with scalar-type properties. To verify the effectiveness of our framework, we adapt these works to be able to generate target samples steered by the number of rings. We acknowledge that, given the exploratory nature of our work, achieving entirely fair comparisons may be unrealistic due to the absence of established baselines.
>
> In fact, these geometry elements are the targets we aim to generate, so evaluating the partition of these elements in the output when they are taken as input may not be as fair.
>
> >> In response to the question, "These geometry elements are the targets we aim to generate, so evaluating the partition of these elements in the output when they are taken as input may not be as fair."
>
> Evaluating the partition of input geometric elements in the output when they are taken as input is critical to evaluating the performance of our framework. The geometric elements are encoded in a way that can steer the generation. Only when the generated output indeed contains the input geometric elements can we verify that our framework can successfully encode the input geometric elements and use the encoded information to steer the generation toward the sparse regions.
>
> As linker design fundamentally is a special case of fragment-based generation, we also conducted a toy experiment on this task to verify the effectiveness of our framework. In particular,  __in this experiment, all algorithms (DiffLinker, LinkerNet, and our method) received the same geometric elements, and we believe this is a fair comparison scenario.__ As shown in Table 6, our experiments revealed that existing approaches exhibit limitations in OOD (data-sparse regions) linker design. The experimental results indicate that existing linker design methods fall short in linking OOD fragments, achieving a validity below 50%. These results demonstrate that our method shows promising performance in fragment linking within the OOD context.
>
> We understand that our explanation might not fully address your comment. You could provide further details or specify the aspects that are unclear. We would be glad to further offer a more targeted explanation.

---

> > ### Comment · Reviewer_kCta · 2024-11-28
> > **Response to Authors**
> >
> > I thank the authors for their detailed response. While my concerns regarding fair comparisons have been partially addressed, some issues remain, particularly regarding the inclusion of certain methods (e.g., EDM) that are **totally** not reasonable for comparison as they are not conditioned on anything. Additionally, I am unclear about the mechanism by which the models take the latent point cloud as input. Specifically, wouldn’t concatenating (latent) node coordinates with node-invariant features break E(3)- and S(N)-equivariance? Could you clarify this aspect rigorously?

---

> > > ### Author Response · Authors · 2024-12-01
> > > **Response to Reviewer kCta**
> > >
> > > # Response to comment 1
> > >
> > > > ... regarding the inclusion of certain methods (e.g., EDM) that are totally not reasonable ...
> > >
> > > We very much appreciate the reviewer's rigor because we are also aware of the significance of providing fairer comparisons to verify the benefits of our work.
> > >
> > > We would like to highlight that our work is the first one to consider the OOD generation under structure shifts. Therefore, we can barely find any baselines that can provide entirely fair comparisons. In response to the need for fair comparisons, comparing with baselines that are not for OOD settings is unexpected. The rationality of using EDM and GeoLDM is that they are representative diffusion-based methods for molecule generation. We intend to reflect on the challenges of using existing generative methods in an OOD setting and to motivate the need for our new design.
> > >
> > > Except for EDM and GeoLDM, we also compared our method with OOD generative models that focus
> > > on property shifts. In addition, we verified the effectiveness of our proposed method by evaluating linker design tasks [1]. As we can achieve SOTA in the OOD generation under the structure shifts, we hope that our work can establish a new milestone on this problem.
> > >
> > >
> > > [1] Igashov, Ilia, et al. "Equivariant 3D-conditional diffusion model for molecular linker design." Nature Machine Intelligence (2024): 1-11.
> > >
> > > # Response to comment 2
> > >
> > > > ... wouldn’t concatenating (latent) node coordinates with node-invariant features break E(3)- and S(N)-equivariance? ...
> > >
> > > We thank the reviewer for the insightful question.
> > >
> > > We would like to highlight that the specific equivariance/invariance required by the molecule generative model is the SE(3)-invariance of the training loss. Under this context, concatenating node coordinates with node-invariant features will __not break SE(3)-invariance of our training loss.__
> > >
> > > Below, we present the detailed proof for clarification.
> > >
> > > The mathematical expression for SE(3)-invariance of our training loss is:
> > >
> > > 1. $\\forall\\mathbf{R}$, $\\mathcal{L}\_{\\textrm{EAAE}}(\\mathbf{x}, \\mathbf{h}, \\mathbf{x}^{f}, \\mathbf{h}^{f})=\\mathcal{L}\_{\\textrm{EAAE}}(\\mathbf{R}\\mathbf{x}, \\mathbf{h},\\mathbf{R}\\mathbf{x}^{f},\\mathbf{h}^{f})$ and
> > > 2. $\\forall\\mathbf{R}, \\mathcal{L}\_{\\textrm{PSDM}}(\\mathbf{z}\_{\\mathbf{x},0},\\mathbf{z}\_{\\mathbf{h},0}, \\mathbf{f})=\\mathcal{L}\_{\\textrm{PSDM}}(\\mathbf{R}\\mathbf{z}\_{\\mathbf{x},0},\\mathbf{z}\_{\\mathbf{h},0}, \\mathbf{f})$.
> > >
> > > We denote $\\mathbf{z}\_{\\mathbf{x},t}$ and $\\mathbf{z}\_{\\mathbf{h},t}$ as $\\mathbf{z}\_{t}$, and $\\mathbf{f}\_{\\mathbf{x}}$ and $\\mathbf{f}\_{\\mathbf{h}}$ as $\\mathbf{f}$ for simplicity, then we have:
> > >
> > > $\\mathcal{L}\_{\\textrm{PSDM}}(\\mathbf{R}\\mathbf{z}\_{\\mathbf{x},0},\\mathbf{z}\_{\\mathbf{h},0}, \\mathbf{f})$
> > >
> > > $=  \\mathbb{E}\_{\\mathcal{G}, \\mathcal{E}\_{\\phi}}\\left[\\sum\_{t=2}^{T}\\mathbb{E}\_{q(\\mathbf{z}\_{t}|\\mathbf{R}\\mathbf{z}\_{0})}[\\text{KL}[q(\\mathbf{z}\_{t-1}|\\mathbf{z}\_{t},\\mathbf{R}\\mathbf{z}\_{0})\\Vert p\_{\\theta}(\\mathbf{z}\_{t-1}|\\mathbf{z}\_{t},\\mathbf{f})]]-\\mathbb{E}\_{q(\\mathbf{z}\_{1}|\\mathbf{R}\\mathbf{z}\_{0})}[p\_{\\theta}(\\mathbf{R}\\mathbf{z}\_{0}|\\mathbf{z}\_{1},\\mathbf{f})] \\right]$
> > >
> > > $=  \\int\_{\\mathcal{G}}\\left[ \\sum\_{t=2}^{T}\\log\\frac{q(\\mathbf{z}\_{t-1}|q(\\mathbf{z}\_{t},\\mathbf{R}\\mathbf{z}\_{0})}{p\_{\\theta}(\\mathbf{z}\_{t-1}|\\mathbf{z}\_{t},\\mathbf{f})}-\\log p\_{\\theta}(\\mathbf{R}\\mathbf{z}\_{0}|\\mathbf{z}\_{1},\\mathbf{f}) \\right]$
> > >
> > > $=  \\int\_{\\mathcal{G}}\\left[ \\sum\_{t=2}^{T}\\log\\frac{q(\\mathbf{R}\\mathbf{R}^{-1}\\mathbf{z}\_{t-1}|q(\\mathbf{R}\\mathbf{R}^{-1}\\mathbf{z}\_{t},\\mathbf{R}\\mathbf{z}\_{0})}{\\mathbf{R}\\mathbf{R}^{-1}p\_{\\theta}(\\mathbf{z}\_{t-1}|\\mathbf{R}\\mathbf{R}^{-1}\\mathbf{z}\_{t},\\mathbf{f})}-\\log p\_{\\theta}(\\mathbf{R}\\mathbf{z}\_{0}|\\mathbf{R}\\mathbf{R}^{-1}\\mathbf{z}\_{1},\\mathbf{f}) \\right] (\\mathbf{R}\\mathbf{R}^{-1}=\\mathbf{I})$
> > >
> > > $=  \\int\_{\\mathcal{G}}\\left[ \\sum\_{t=2}^{T}\\log\\frac{q(\\mathbf{R}^{-1}\\mathbf{z}\_{t-1}|q(\\mathbf{R}^{-1}\\mathbf{z}\_{t},\\mathbf{z}\_{0})}{\\mathbf{R}^{-1}p\_{\\theta}(\\mathbf{z}\_{t-1}|\\mathbf{R}^{-1}\\mathbf{z}\_{t},\\mathbf{f})}-\\log p\_{\\theta}(\\mathbf{z}\_{0}|\\mathbf{R}^{-1}\\mathbf{z}\_{1},\\mathbf{f}) \\right] (SE(3)\\ \\text{of}\\ \\mathbf{z}\_{t})$
> > >
> > > $=  \\mathbb{E}\_{\\mathcal{G}, \\mathcal{E}\_{\\phi}} \\left[ \\sum\_{t=2}^{T}\\log\\frac{q(\\mathbf{j}\_{t-1}|q(\\mathbf{j}\_{t},\\mathbf{z}\_{0})}{\\mathbf{R}^{-1}p\_{\\theta}(\\mathbf{z}\_{t-1}|\\mathbf{j}\_{t},\\mathbf{f})}-\\log p\_{\\theta}(\\mathbf{z}\_{0}|\\mathbf{j}\_{1},\\mathbf{f}) \\right] (\\text{Let}\\ \\mathbf{j}\_{t}=\\mathbf{R}^{-1}\\mathbf{z}\_{t})$
> > >
> > > $=  \\mathcal{L}\_{\\text{PSDM}}(\\mathbf{z}\_{\\mathbf{x},0},\\mathbf{z}\_{\\mathbf{h},0}, \\mathbf{f})$
> > >
> > > The complete proof is presented in Appendix D on Page 18.

---

### Official Review · Reviewer_wmeZ · 2024-10-28

**Soundness:** 2
**Presentation:** 1
**Contribution:** 1
**Rating:** 3
**Confidence:** 3

**Summary:**

The paper addresses the problem of sampling molecules given a sub-structure (fragment). The outlined approach could be an important tool in computer-aided drug-design, in particular in fragment based discovery.

The approach is formulated as a denoising diffusion probabilistic model for sampling molecules conditioned on a learned embedding of a scaffold (confusingly named a 'physical prior') -- but essentially is a learned generative 'unmasking' or 'inpainting' task. The embedding is achieved by using an 'asymmetric equivariant auto encoder,' as the noise-prediction model.

Minor comments:
In 2.2, line 160, a weight is defined but it is not used anywhere.

**Strengths:**

a. Good performance compared to baselines.

**Weaknesses:**

a. Overly complicated and pompous presentation considering the method. It is unclear whether authors are trying to obscure the simplicity in the method by invoking overly convoluted language.  For example use "physical prior" to denote a learned embedding used to guide/steer generation.
b. Proofs on the equivariance are presented as new, yet are simple (trivial?) extensions of well known proofs in the field, and there are no references to the original work e.g. Kohler et al ICML 2020 and others. Can the authors perhaps clarify the connection to this previous work and highlight how what they present is different?
c. While the proposed method beats a range of baselines, it is unclear whether results are good enough to be useful in a practical setting. The dataset which would be a useful benchmark for a method like this, the molecular stability should be at least 50%. On QM9 which is arguably not a helpful benchmark for this task, e.g. fragment based drug design, the method barely reaches this level.

**Questions:**

1. I do not understand the statement that molecular stability is close to 0% on the GEOM-DRUG baseline. It is reported for most standard molecular generation tasks, benchmarks you compare against have ~80-92% AS for unconditional sampling. Are you referring to the 'OOD' generation task? That seems to be the case since the atom stability is very low (Table 3), so low in fact that it is unlikely that any molecules that are generated are even valid. It seems like the comparison is nearly meaningless. For QM9 the results are better, but still far from being utile in a practical setting, since QM9 is unlikely to be used for fragment based drug discovery.

---

> ### Author Response · Authors · 2024-11-24
> **Response to Reviewer wmeZ**
>
> We thank you for your time and effort for reviewing our works.
>
> # Response to minor comments
> > In 2.2, line 160, a weight is defined, but it is not used anywhere.
>
> We appreciate the reviewer's careful review. The weight defined here is used in the diffusion loss, as illustrated in Eq. (17) on page 14.
>
> # Response to Weakness a
> > a. Overly complicated and pompous presentation considering the method ...
>
> We appreciate the reviewer’s comment. Technically, we concur that the proposed method utilizes a learned embedding to steer the generation process. Conceptually, this embedding is derived from the sub-structures of a molecule, such as scaffolds and ring structures. Those sub-structures typically are stable backbones for designing bespoke properties, which could be treated as critical physical prior information. We would like to clarify that our intention is not to obscure the simplicity of the method; instead, we aim to convey our ideas accurately and transparently.
>
>
>
> # Response to Weakness b
> > b. Proofs on the equivariance are presented as new, yet are simple (trivial?) extensions of well-known proofs in the field ...
>
> We appreciate the reviewer’s recognition of our proofs as new and simple. We acknowledge that our proofs are related to but different from existing works. The work by Kohler et al. established that a transformation is H-invariant under certain constraints. In contrast, __our proof specifically addresses SE(3)-invariance__. Furthermore, the main difference is that our framework's training loss satisfies invariance when conditioned on equivariant representations. This aspect has not been considered or proven before, serving as the core theoretical foundation of our work. For clarity, we have included a discussion of previous works in the revised manuscript. We added citations to this work and supplemented extra discussion in the revised manuscript.
>
> # Response to Weakness c
> > c. While the proposed method beats a range of baselines ...
>
> In fact, we agree with the reviewer that all work on molecule generation, including ours, is not practical enough. Among all baselines, ours has achieved SOTA performance in commonly used benchmarking datasets, including the GEOM-DRUG and QM9 datasets, as well as real-world applications in linker design. Our work can help push forward the frontier in this area.
>
> Moreover, we are aware that the complete QM9 dataset may not serve as an ideal benchmark for molecular generation, as it consists of a comprehensive enumeration of elements H, C, N, O, and F. To address this issue, we have divided the dataset based on data scarcity, specifically focusing on the frequency of scaffolds and ring structures. Thus, while QM9 may not be suitable for training a real-world molecular generator, it remains an effective benchmark for comparing different algorithms.
>
> # Response to Question 1
> > I do not understand the statement that molecular stability is close to 0% on the GEOM-DRUG baseline ...
>
> We appreciate the reviewer’s astute observation. The reported 0\% molecular stability on the GEOM-DRUG dataset indeed reflects the results of the methods evaluated in the OOD generation task, highlighting both the challenges of the dataset and the limitations of existing OOD generation methods. While our proposed method has not yet achieved optimal performance, we believe that the transition from zero to 11.4\% represents a significant improvement. We acknowledge the difficulties associated with this task and plan to investigate larger molecular datasets in future work.

---

> > ### Comment · Reviewer_wmeZ · 2024-11-24
> >
> > Thanks for addressing my comments. I will maintain my score.

---

> > > ### Author Response · Authors · 2024-11-25
> > > **Response to Reviewer wmeZ**
> > >
> > > We thank you for identifying our strengths and acknowledging that __we have addressed all your concerns__. In addition to your recognition, we summarized other reviewers’ recognition of our contributions point by point below.
> > >
> > > 1. The Reviewer RMWa found that our target issue is challenging and underexplored, the analysis is comprehensive, and the metrics are beneficial for future works. ([link](https://openreview.net/forum?id=an3kPpce6b&noteId=WkkXTBZA2R))
> > > 2. The Reviewer eRCJ found that our work is interesting, opens up new exploration and opportunities for AI for drug discovery, and our paper is well-written and clear. ([link](https://openreview.net/forum?id=an3kPpce6b&noteId=y1gBT1oTs7))
> > > 3. The Reviewer kCta recognized the value and practical importance of our research, found our results promising, and commended the organization of our code. ([link](https://openreview.net/forum?id=an3kPpce6b&noteId=gAeT1MxNhD))
> > >
> > > We sincerely invite the reviewer to reassess our paper. If you have any further concerns, we will be glad to respond.

---

> > > > ### Comment · Reviewer_wmeZ · 2024-11-25
> > > >
> > > > Thanks for summarizing the other reviewers positions. However, my main concern is about the practicality of the approach as it stands now, consequently I remain firm in my decision to maintain my score.

---

> > > > > ### Author Response · Authors · 2024-12-01
> > > > > **Response to Reviewer wmeZ**
> > > > >
> > > > > # Response to the practicality of the approach
> > > > >
> > > > > > Thanks for summarizing the other reviewers positions. However, my main concern is about the practicality of the approach as it stands now, consequently I remain firm in my decision to maintain my score.
> > > > >
> > > > > We understand and appreciate the reviewer's concern regarding practicality, a critical aspect of AI for drug discovery. We also aim to propose high-impact works that can be used in the real world.
> > > > >
> > > > > Thus far, due to the open challenges of AI for drug design, most existing AI-based molecule generative frameworks remain distant from practical applications. Among the current challenging issues, we specifically focus on dealing with the difficulty due to data scarcity [1]. In particular, to address the data scarcity issue in AI for drug design, we proposed a novel molecular generator that can steer the generation towards data-sparse regions given data-abundant regions; our proposed generator can produce molecules that are not in the training data, which is of critical importance for drug discovery [2].
> > > > >
> > > > > We are aware of the practicality verification of our framework, which is exactly the same as the concern pointed out by the reviewer. For prototype practicality verification, we conducted experiments within fragment-based drug design (including the linker design task), which is a mainstream real-world drug design method [3]. We hope our clarification on these points can alleviate your concern.
> > > > >
> > > > > [1] Klarner, Leo, et al. "Context-guided diffusion for out-of-distribution molecular and protein design." ICML: (2024).
> > > > >
> > > > > [2] Walters, W. Patrick, and Mark Murcko. "Assessing the impact of generative AI on medicinal chemistry." Nature biotechnology 38.2 (2020): 143-145.
> > > > >
> > > > > [3] Murray, Christopher W., and David C. Rees. "The rise of fragment-based drug discovery." Nature chemistry 1.3 (2009): 187-192.

---

### Official Review · Reviewer_eRCJ · 2024-11-03

**Soundness:** 3
**Presentation:** 3
**Contribution:** 3
**Rating:** 6
**Confidence:** 4

**Summary:**

This paper presents the Geometric Out-of-Distribution Diffusion Model (GODD), a framework for generating 3D molecular structures in data-sparse regions by framing the challenge as an out-of-distribution (OOD) generation problem. It introduces an asymmetric encoder-decoder architecture that captures distributional physical priors from molecular fragments, enabling the model to generalize to unseen structural variations without additional training on OOD data. By leveraging these physical priors to guide the denoising process, the framework significantly improves the success rate of generating valid and chemically relevant molecules, demonstrating its effectiveness in fragment-based drug design tasks and ensuring the preservation of geometric properties through SE(3)-equivariance.

**Strengths:**

1. The paper introduces a method to solve the OOD generalizability problem with current SOTA diffusion models, enhancing the generation of valid 3D molecular structures in data-sparse regions.
2. The paper introduces geometric OOD and formulate distributional physical priors from molecular fragments, enabling effective capture and utilization of structural information to guide the generation process.
3. The paper is well-written, featuring clear presentation and thorough theoretical analysis that demonstrates the SE(3)-equivariance of the physical priors, ensuring geometric consistency in the generated molecules.

Overall I think this is an interesting work and open up new exploration and opportunities for AI for drug discover in terms of OOD and generalization ability.

**Weaknesses:**

1. The practical implications and real-world application scenario in drug discovery of geometric OOD are not clear, so the authors should explain more about the use case of their framework in this context.
2. The paper primarily focuses on OOD generation related to shifts on molecule fragments, which could be limited to scenarios where the general structure of fragments is not clear.

**Questions:**

Please see weakness part.

---

> ### Author Response · Authors · 2024-11-24
> **Response to Reviewer eRCJ**
>
> We are grateful that the reviewer acknowledges the strengths of our work. We are glad that the reviewer found that __our work is interesting and opens up new exploration and opportunities for AI for drug discovery__ and __our paper is well-written and clear__. We address the reviewer's concerns below.
>
> # Response to Weakness 1
> > The practical implications and real-world application scenario ...
>
> We appreciate the reviewer’s comment. In drug discovery, OOD generation is crucial due to challenges related to data scarcity and the need for novel compounds. First, training a molecular generator typically requires a substantial amount of molecular data. However, for specific tasks, particularly those related to emerging diseases, available data may be insufficient [1]. Second, an effective molecular generator should avoid replicating the training data [2]; OOD generation enables models to create molecular structures that were not present in the training dataset.
>
> [1] Klarner, Leo, et al. "Context-guided diffusion for out-of-distribution molecular and protein design." ICML: (2024).
>
> [2] Walters, W. Patrick, and Mark Murcko. "Assessing the impact of generative AI on medicinal chemistry." Nature biotechnology 38.2 (2020): 143-145.
>
> # Response to Weakness 2
> > The paper primarily focuses on OOD generation related to shifts on molecule fragments ...
>
> We appreciate the reviewer’s insightful comment. We acknowledge the limitation you point out and have addressed it in the limitation section of our paper. In our future work, we will actively explore methods to extract the general structure of fragments that represent distribution shifts but are not given, thereby enhancing the generalizability of our approach.
>
> We also would like to clarify the significance of our work. Scaffolds/fragments are crucial in drug design as they provide a stable backbone that can be decorated with different functional groups to optimize the molecule's properties. Providing pre-defined scaffolds/fragments helps explore the chemical space efficiently and generate novel compounds with desired characteristics. Therefore, molecule generation given pre-defined scaffolds/fragments is an important task with wide applications.

---

> > ### Comment · Reviewer_eRCJ · 2024-12-01
> > **Response to Authors**
> >
> > Thanks the authors for their rebuttal. I have no further questions.

---

### Official Review · Reviewer_RMWa · 2024-11-04

**Soundness:** 4
**Presentation:** 4
**Contribution:** 3
**Rating:** 6
**Confidence:** 4

**Summary:**

To address the generation of OOD molecules with rare scaffolds, this work proposes a diffusion framework conditioned on physical priors from molecular fragments. The model comprises an equivariant autoencoder that encodes the 3D molecular fragment graph into equivariant and invariant latent representations as physical priors, and a diffusion model that generates molecules conditioned on the priors. The authors test the method on OOD ring and scaffold structures and show superior performance than existing methods.

**Strengths:**

1. This work focuses on the challenging and poorly explored problem of generating 3D molecules from the low-data domains. The proposed model shows promising results across several tasks.
2. The authors provide comprehensive analysis on the generation results.
3. The authors also propose multiple useful evaluation metrics that could facilitate future studies on the subject.

**Weaknesses:**

1. From the presented generation examples, the scaffold seems to constitute a large proportion of the molecule. Thus, it is possible that the model actually learns to directly modify the given scaffold locally (as the scaffold is almost a valid molecule already) instead of really transferring the knowledge of relations between scaffolds and the full molecule structure from the data rich domain. Some additional analysis are needed to prove the claim (see Questions).
2. The actual application of the method is somewhat limited as a pre-defined scaffold structure needs to be provided to the model.

**Questions:**

Major:
1. For molecules with a large number of rings, the variations of ring structure is exponentially larger. How is the input scaffold selected for the generation of molecules >10 rings?
2. Since the scaffold information is provided as a condition, not explicitly enforced, it is possible that the scaffold is edited (e.g. slight changes in atom types and coordinates) during the generation due to the stochastic nature of DDPM. From Fig 11, it seems to be quite common. How is the proportion and coverage calculated with those cases?
3. For linker design tasks, the initial structures are two molecular fragments. In this scenario, how is the scaffold defined and provided to the diffusion model?
4. Following the previous question, if the model could work with disconnected scaffold in linker design, would it also work with incomplete scaffolds? (e.g. providing an increasing proportion of the fragment to the diffusion model).
5. As mentioned in "limitations", a known scaffold structure needs to be provided. Is it possible to generalize it to randomly generated OOD scaffold structures?

Minor:
1. The authors name the prior as “physical”, while it is actually learned only from the 3D structure of the fragments. What is “physical” about it?
2. For molecules with a large number of rings, the variations of ring structure is exponentially larger. How is the input scaffold selected for the generation of molecules >10 rings?
3. As $f_h$ only has 1 dimension per atom, is the representation expressive enough? Would the model benefit from higher dimensionality?
4. How is the atom number determined when sampling $z_{x,T}$, $z_{h,T}$?

---

> ### Author Response · Authors · 2024-11-24
> **Response to Reviewer RMWa (1/2)**
>
> We are pleased that the reviewer acknowledges the strengths of our work. We appreciate that the reviewer found that our target issue is __challenging and underexplored__, __results are promising across various tasks__, __analysis is comprehensive__, and __the metrics are beneficial for future works__. We address the reviewer's concerns as follows.
>
> # Response to Weakness 1 and Major Question 2
> > W1. From the presented generation examples, the scaffold seems to constitute a large proportion of the molecule...
>
> > Major Q2. Since the scaffold information is provided as a condition ...
>
> __In response to Weakness 1 and Major Question 2__: We concur that the scaffold constitutes a significant portion of the molecule, which could be modified to create various derivatives with different properties. We would like to clarify that our diffusion model is trained in a way that it can sample the targeted molecules during the generation process rather than generate new samples based on modifications over any scaffolds. As verified with our experiments, our framework can generate samples containing specific scaffolds, which are rare in the dataset, as shown in Tables 4 and 5. Our experiments could be strong evidence showing the transferability from the dense region to the sparse regions.
>
>
> __In response to the Fig. 11 and the calculation of proportion and coverage.__ There are indeed slight alterations in the output compared to the input ring structure depicted in Fig. 11. But these outputs are our desired molecules as long as they contain the target ring number in the context of ring-structure generation tasks. Specifically, the proportion is defined as the percentage of generated molecules that contain the desired ring numbers among the valid samples. However, the coverage metric is not applicable in this context, as each ring structure represents a unique target distribution. Furthermore, in scaffold-based generation tasks, both coverage and proportion are calculated when the output molecule accurately incorporates the desired scaffold.
>
>
> # Response to Weakness 2
> > W2. The actual application of the method is somewhat limited as a pre-defined scaffold structure needs to be provided to the model...
>
> __In response to Weakness 2__: Scaffolds are crucial in drug design as they provide a stable backbone that can be decorated with different functional groups to optimize the molecule's properties. Providing pre-defined scaffolds helps explore the chemical space efficiently and generate novel compounds with desired characteristics. Therefore, molecule generation given pre-defined sub-structures is an important task with wide applications.
>
>
>
> # Response to Major Question 1 and Minor Question 2
> > Q. For molecules with a large number of rings ...
>
> __In response to Major Question 1 and Minor Question 2__: We began by analyzing the dataset to identify and filter out molecules containing more than ten rings, which are in sparse regions (all fewer than 100 samples). We then extracted the ring structures from these selected molecules; the extracted ring structures are used as input for generating new molecules with more than ten rings. This process ensures that these molecules in sparse regions are not provided for training.
>
>
>
> # Response to Major Question 3
> > Major Q3. For linker design tasks, the initial structures are two molecular fragments ...
>
> In linker design tasks, the initial structures consist of multiple molecular fragments, all fed into the equivariant autoencoder. The intuition of using an equivariant autoencoder is to learn expressive representations of the fragments, which are used to steer the linking process toward OOD generation. We did not define scaffolds for this task. Instead, we directly use the molecular fragments as the input of the autoencoder.

---

> ### Author Response · Authors · 2024-11-24
> **Response to Reviewer RMWa (2/2)**
>
> # Response to Major Question 4
> > Major Q4. Following the previous question ...
>
> We sincerely appreciate the reviewer's insightful comment. First, the model indeed could work with disconnected scaffolds. Second, we can conjecture that the proposed method can also work with incomplete scaffolds as incomplete scaffolds and complete scaffolds share the same representation methods. For the reviewer's suggested task, we would like to explore relevant real-world tasks that provide an increasing proportion of the fragments and extend our frameworks as our future work.
>
> # Response to Major Question 5
> > Major Q5. As mentioned in "limitations" ...
>
> It is possible. As mentioned in response to Major Question 4, our framework is agnostic to the detailed structure of the input scaffold as long as it can be presented in geometric point clouds.
>
> # Response to Minor Question 1
> > Minor Q1. The authors name the prior as ''physical''...
>
> We appreciate the reviewer's insight regarding the prior being derived from the 3D structure of the fragments/scaffolds. Generally, scaffolds are crucial in drug design as they provide a stable backbone that can be decorated with different functional groups to optimize the molecule's properties. In this context, scaffolds or input 3D structures are critical physical information. Specifically, this physical knowledge serves as a prior in guiding OOD generation in data-sparse regions; thereby, we use physical priors to denote these input 3D structures.
>
> # Response to Minor Question 3
> > Minor Q3. As $f_{h}$ only has 1 dimension per atom ...
>
> Thank you for your insightful question. Regarding the latent dimensionality of $\mathbf{f}_{\mathbf{h}}$, we adhere to previous studies on the latent representation of geometric point clouds. Research indicates that a smaller latent dimensionality is beneficial for effectively representing geometric information. In response to your concern, we conducted experiments on this parameter and presented the results below.
>
> | Metrics | Metrics | P (%) | in   |  Distribution | Generation | P (%) | in   | OOD  | Generation |      | AS  (%)                 | MS (%) | V  (%) | N (%) | S (%) |
> | ------- | ------- | ----- | ---- | ------------- | ---------- | ----- | ---- | ---- | ---------- | ---- | ----------------------- | ------ | ------ | ----- | ----- |
> | Methods | \# Ring | 0     | 1    | 2             | 3          | 4     | 5    | 6    | 7          | 8    | Averaged | over |  9 | Domains |
> | Dataset | QM9     | 10.2  | 39.3 | 27.6          | 15.1       | 4.4   | 2.7  | 0.6  | 0.2        | 0.0  | 99.0                    | 95.2   | 97.7   | \-    | \-    |
> | GOOD    | $k=16$    | 96.2  | 96.3 | 95.5          | 93.9       | 89.2  | 86.4 | 75.8 | 85.1       | 80.3 | 80.2                    | 52.1   | 75.1   | 67.8  | 39.1  |
> | GOOD    | $k=8$     | 97.4  | 97.4 | 96.7          | 95.1       | 90.2  | 87.5 | 76.8 | 86.1       | 81.3 | 81.1                    | 52.7   | 76.0   | 68.6  | 39.6  |
> | GOOD    | $k=4$     | 98.6  | 98.6 | 97.9          | 96.4       | __92.6__  | 88.6 | 77.8 | 87.2       | 82.0 | 82.1                    | 53.4   | 76.9   | 69.5  | 40.0  |
> | GOOD    | $k=1$     | __99.9__  | __99.8__ | __99.1__          | __97.6__ | 92.5  | __89.7__ | __78.7__ | __88.2__       | __82.1__ | __83.1__ | __54.0__   | __77.9__   | __70.3__  | __40.5__  |
>
> The experimental results demonstrate that the dimensionality of the latent atomic features $\mathbf{f}_{\mathbf{h}}$ has a minimal impact on performance, with $k=1$ yielding the best results. This finding is consistent with observations in GeoLDM [1], which also confirmed that a small latent dimensionality for atomic features remains representative of reconstructing the whole molecule.
>
> [1] Xu, Minkai, et al. "Geometric latent diffusion models for 3d molecule generation." International Conference on Machine Learning. PMLR, 2023.
>
> # Response to Minor Question 4
> > Minor Q4. How is the atom number determined when sampling $z_{x,T}$, $z_{h,T}$ ?
>
> We determine the atom number based on established methodologies from the literature. The number of atoms is sampled from the distribution derived from the entire dataset. Additionally, we exclude any sizes that fall below the size of input fragments.

---

> > ### Comment · Reviewer_RMWa · 2024-11-24
> >
> > I appreciate the authors' efforts in addressing my concerns and confusions. I have no other questions for now.

---

### Author Response · Authors · 2024-11-24
**General Response**

We sincerely appreciate the time and efforts of the reviewers in providing their valuable feedback.

The key points of our rebuttal can be summarized as follows:

1. Provide extensive discussion and clarifications of the following aspects

> 1.1. The significance of the problem ([link](https://openreview.net/forum?id=an3kPpce6b&noteId=YH0cRXiavA), [link](https://openreview.net/forum?id=an3kPpce6b&noteId=xCAtHuheXS))

> 1.2 The rationality of using "physical prior" ([link](https://openreview.net/forum?id=an3kPpce6b&noteId=tX8Qf6bxZq), [link](https://openreview.net/forum?id=an3kPpce6b&noteId=zTrP4I7r3k))

> 1.3 The proofs on the equivariance ([link](https://openreview.net/forum?id=an3kPpce6b&noteId=zTrP4I7r3k))

> 1.4 The implication of our performance in practical settings ([link](https://openreview.net/forum?id=an3kPpce6b&noteId=zTrP4I7r3k))

> 1.5 Ambiguous statements of the notations and figures ([link](https://openreview.net/forum?id=an3kPpce6b&noteId=7CVziWpBDL))

> 1.6 Fairness justification of our experimental comparisons ([link](https://openreview.net/forum?id=an3kPpce6b&noteId=0xxhBxWL3f))

2. Conduct additional experiments on the following aspects:

> 2.1 The effectiveness of different dimensions for $\\mathbf{f}\_{\\mathbf{h}}$ ([link](https://openreview.net/forum?id=an3kPpce6b&noteId=tX8Qf6bxZq))

---

### Meta-Review · Area_Chair_TZnY · 2024-12-20

**Metareview:**

This paper introduce GODD, a diffusion-based generative framework that generalizes from data-abundant to data-scarce distributions using an equivariant asymmetric autoencoder to capture physical priors. This allows effective generation without training on sparse data. Experiments show a 65.6% improvement in success rate, outperforming baselines in molecular validity, uniqueness, and novelty. GODD also adapts well to fragment-based drug design tasks.

**Strengths:**

1. The task of generating out-of-distribution (OOD) molecules is valuable, particularly for applications like drug design.
2. The paper introduces the concept of geometric OOD and formulates distributional physical priors based on molecular fragments.
3. The authors provide a thorough analysis of the generation results.

**Weaknesses:**

1. The applicability of the method is somewhat constrained, as it requires a pre-defined scaffold structure for the model.
2. The experimental comparisons may be biased and not fully representative.
3. The reliability of experiments on GEOM-DRUG needs improvement, as all compared methods yield zero results.

**Overall:** This is a borderline paper. While the proposed OOD generation task holds promise, the current experimental validation is unconvincing, which limits the approach's practicality. Therefore, I recommend rejection.

**Additional Comments On Reviewer Discussion:**

In the rebuttal period, the authors effectively addressed the concerns raised by Reviewers RMWa and eRCJ, who primarily questioned the limitation of the method, which requires a pre-defined scaffold structure. Reviewers wmeZ and kCta expressed concerns about the fairness and reliability of the comparisons with unconditional baselines. Despite the authors' feedback, Reviewer wmeZ maintains a negative opinion, particularly regarding the practicality of the approach. Given that the major concerns raised by the reviewers remain unresolved, this paper is not yet ready for publication in its current version.

---

### Decision · Program_Chairs · 2025-01-22

Reject